# Eco-sustainable magnetoresistive sensors towards disposable magnetoelectronics

Lin Guo [1], Rui Xu [1]✉, Proloy Taran Das[1], Eduardo Sergio Oliveros-Mata [1], Xuan Peng[2], Oleksandr V. Pylypovskyi [1], René Hübner [1], Fabian Ganss [1], Xiaotao Wang[1], Yi Li[1], Sebastian Gepp[3], Yevhen Zabila[1], Xilai Bao[4], Shengbin Li [4], Qihao Zhang [1], Igor Veremchuk[1], Željko Janićijević [2], Larysa Baraban [2,5], Clemens Voigt[6], Sindy Mosch[6], Oliver Gutfleisch [7], Run-Wei Li[4,8] & Denys Makarov [1]✉

This work presents a holistic integration of environmental sustainability and enhanced sensing performance throughout the full lifecycle of magnetoresistive sensors. Utilizing industry-scale screen-printing techniques combined with eco-friendly inks (formulated from engineered $Fe/Fe_3O_4$ core-shell magnetic microparticles, bioderived polymeric binders, and water solvent), the fabrication process avoids harsh treatments and hazardous chemicals. The resulting sensors, constructed entirely from naturally sourced materials, inherently exhibit biocompatibility, biodegradability, and environmentally benign recyclability. These properties collectively demonstrate key attributes for a sustainable life cycle. Through rational engineering of the $Fe/Fe_3O_4$ core-shell structure particles, two synergistic mechanisms are activated within the composite: spin-dependent hopping across $Fe_3O_4$ shell grain boundaries and in situ magnetic flux concentration induced by Fe cores, thereby yielding an order-of-magnitude enhancement in low-field sensitivity relative to sputtered Fe film and printed $Fe_3O_4$ particle-based counterparts, resulting in a higher magnetoresistance ratio at 10 mT relative to all printed magnetoresistive sensors reported previously. The convergence of eco-sustainability and high performance enables previously unattainable disposable magnetoelectronics, unlocking new opportunities for environmentally responsible and user-safe transient electronics and Internet of Things (IoT) applications.

The rapid advancement of digital technologies has profoundly reshaped contemporary society, enabling unprecedented levels of connection, interaction, and automation. However, the widespread incorporation of electronic devices has also introduced substantial environmental challenges. Central to the operation of these electronic systems are diverse sensor technologies, which serve as essential transducers between physical phenomena and digital information. Among them, magnetoresistive sensors, capable of environment-

[1]Helmholtz-Zentrum Dresden-Rossendorf e.V., Institute of Ion Beam Physics and Materials Research, Dresden, Germany. [2]Helmholtz-Zentrum Dresden-Rossendorf e. V., Institute of Radiopharmaceutical Cancer Research, Dresden, Germany. [3]Freudenberg Siebdruck GmbH, Dresden, Germany. [4]CAS Key Laboratory of Magnetic Materials and Devices, Ningbo Institute of Materials Technology and Engineering, Chinese Academy of Sciences, Ningbo, China. [5]Technische Universität Dresden , Else Kröner Fresenius Center for Digital Health (EKFZ), Faculty of Medicine Carl Gustav Carus, Dresden, Germany. [6]Fraunhofer Institute for Ceramic Technologies and Systems IKTS, Dresden, Germany. [7]Institute of Materials Science, Technical University of Darmstadt, Darmstadt, Germany. [8]Eastern Institute of Technology, Ningbo, China. ✉e-mail: r.xu@hzdr.de; d.makarov@hzdr.de

resilient non-contact detection of magnetic fields and consequently relative motion, have been extensively used across a broad spectrum of applications, ranging from environmental monitoring and industrial automation to biomedical diagnostics and the Internet of Things (IoT)[1–5]. Owing to their functional versatility and robustness, over 100 billion magnetoresistive sensor units were deployed globally in 2022, with market projections estimating that this number could exceed 1 trillion units by 2030[6]. The massive scale of production and deployment raises pressing sustainability concerns, necessitating urgent scientific and technological attention.

In alignment with the United Nations Sustainable Development Goals (SDGs)[7], which call for environmentally responsible design, production, and disposal of electronic products, there has been growing momentum toward the development of sustainable electronics. For example, significant efforts have been taken to develop biodegradable electronics[8–11] that safely decompose at the end of their functional life without releasing hazardous substances into the environment. Recent advances in biocompatible materials have enabled a variety of biodegradable sensors (e.g., temperature, opticals, pressure/strain, chemical, humidity, acoustic)[12–14]. Despite these advances, the majority of magnetoresistive sensors still rely heavily on Co and Ni, both of which are classified as hazardous by the Globally Harmonized System of Classification and Labelling of Chemicals[15,16]. Their widespread usage and improper disposal pose serious ecological and health risks. Moreover, achieving high magnetoresistive performance necessitates precise control over spin-dependent effects, which are generally realized through high-vacuum material deposition and subtractive lithographic patterning, i.e., processes associated with high energy consumption and material waste.

Printable magnetoresistive sensors based on functional inks (composed of magnetic fillers and polymeric binder solutions) present a promising avenue to address the above challenges, thanks to their compatibility with material- and energy-efficient manufacturing techniques[17–20]. However, most reported printable magnetoresistive sensors that demonstrate acceptable levels of magnetoresistance still rely on Ni- or Co-based fillers[21–31] (Supplementary Table 1). In addition, the ink formulations and printing processes often involve hazardous organic chemicals and solvents, further undermining their sustainability. As a greener alternative, ferromagnetic Fe has attracted attention. Nonetheless, Fe-based sensors are fundamentally limited by their intrinsically low magnetoresistance (<0.5%)[32,33] (Supplementary Table 2). In contrast, $Fe_3O_4$-film based sensors can exhibit higher magnetoresistance (1–10%), but only under strong magnetic fields (>1 T)[34–37] (Supplementary Table 3). This poses a significant limitation for real-world applications, particularly in wearable and consumer electronics, where continuous exposure to high magnetic fields, exceeding the 40 mT threshold recommended by the World Health Organization (WHO)[38], is undesirable. Even worse, when these materials are processed via printing techniques, inevitable structural heterogeneities, poor inter-filler connectivity, and reduced magnetic domain alignment further suppress magnetoresistive response. In addition, in contrary to thin-film technology that affords nanoscale control, printing techniques typically feature dimensions on the order of tens or hundreds of micrometers. This scale mismatch, combined with the inherently high resistivity of $Fe_3O_4$, poses a vital challenge for realizing printed $Fe_3O_4$-based sensors with practical functionality due to substantial electrical resistivity. Collectively, these challenges highlight an urgent need for the development of new material systems and fabrication strategies that can bridge the sustainability and functionality in magnetoresistive sensors.

In this work, we integrate environmental sustainability considerations across key lifecycle stages of printable magnetoresistive sensors while equipping them with enhanced low-field sensing performance. To this end, we develop an eco-friendly ink formulated with naturally abundant Fe as the primary sensing composition, bioderived

polymers as binders, and water as the solvent. This ink is compatible with industrial-scale screen printing techniques, enabling cost-effective high-throughput sensor fabrication, as evidenced by 5 × 8 sensor arrays printed on A4-sized papers. The resulting sensors exhibit excellent biocompatibility, with minimal cytotoxicity observed in murine fibroblast cells in vitro. After end-of-life, the sensors can be disposed of through both controllable biodegradation and green recyclability, ensuring eco-safe disposal. To endow the device-applicable performance, we rationally engineer the $Fe/Fe_3O_4$ core-shell magnetoresistive microparticles and their matrix within the composite, leveraging two synergistic mechanisms: (i) spin-dependent hopping across the $Fe_3O_4$ grain boundaries at the interface between adjacent shells, enhancing the magnetoresistance response; (ii) an in situ magnetic flux guiding effect induced by the Fe core, thereby amplifying the local field at the interface of adjacent $Fe_3O_4$ shell-shell grain boundaries. Together, the two effects yield remarkable improvements in low-field magnetoresistance and sensitivity. Specifically, two parameters are enhanced by factors of 27.5 and 8, respectively, compared to Fe films, and by 3.4 and 20 times, respectively, relative to printed $Fe_3O_4$-based counterparts. Further, the $Fe/Fe_3O_4$ core-shell structures offer an additional advantage: the Fe core serves as a highly conductive pathway that electrically connects the $Fe_3O_4$ sensing domains, resulting in resistivity reduction by over two orders of magnitude compared to that of printed $Fe3O_4$-based composite, facilitating seamless integration into electronic systems. The simultaneous realization of a sustainable lifespan and superior low-field sensitivity opens new opportunities for disposable magnetoelectronics, with potential applications in smart packaging, food/medicine monitoring, and plant/human wearables and implantables.

## Results
### Fully green printable fabrication of high-performance biodegradable magnetoresistive sensors
Figure 1a schematically illustrates the formulation strategy of the functional ink, of which all components are free from scarce resources or hazardous substances. The engineered $Fe/Fe_3O_4$ core-shell structure endows the ink with magnetoresistive response, and the constituent Fe element is not only abundant in the crust but also plays an essential role in various biological processes[39]. Sodium carboxymethyl cellulose (NaCMC), derived from plant-based cellulose, is a renewable non-toxic polymer, and has been extensively employed in pharmaceutical and tissue engineering applications. Replacing conventional volatile organic solvents with water eliminates ecological hazards during ink preparation while promoting a safer printing process. Through systematic engineering of these constituent materials, the formulated ink exhibits high compatibility with standard screen-printing techniques (Fig. 1b and Supplementary Fig. 1). Rheological measurements further corroborate the screen-printing suitability of the selected ink, featuring the suitable viscosity for screen printing and pronounced shear-thinning under increasing shear rate (Supplementary Fig. 2). After printing, the aqueous solution in the ink evaporates at ambient conditions, causing substantial volume shrinkage. This simple yet effective method induces internal stress, which tightly compresses the microparticles to establish electrical percolation pathways. The solidified NaCMC immobilizes the magnetoresistive microparticles, thereby forming the robust building block of the sensing device. The printing workflow is free from any harsh post-treatments, reducing associated environmental burdens. As a demonstration of scalability, 5 × 8 sensor arrays are fabricated on A4-sized paper substrates (Fig. 1b, c, Supplementary Fig. 1 and Supplementary Movie 1). The printed sensors also exhibit robust device-to-device performance reproducibility (Supplementary Fig. 3 and Supplementary Table 4). This process highlights the feasibility of low-cost high-throughput production of magnetoresistive sensors using a fully green method.

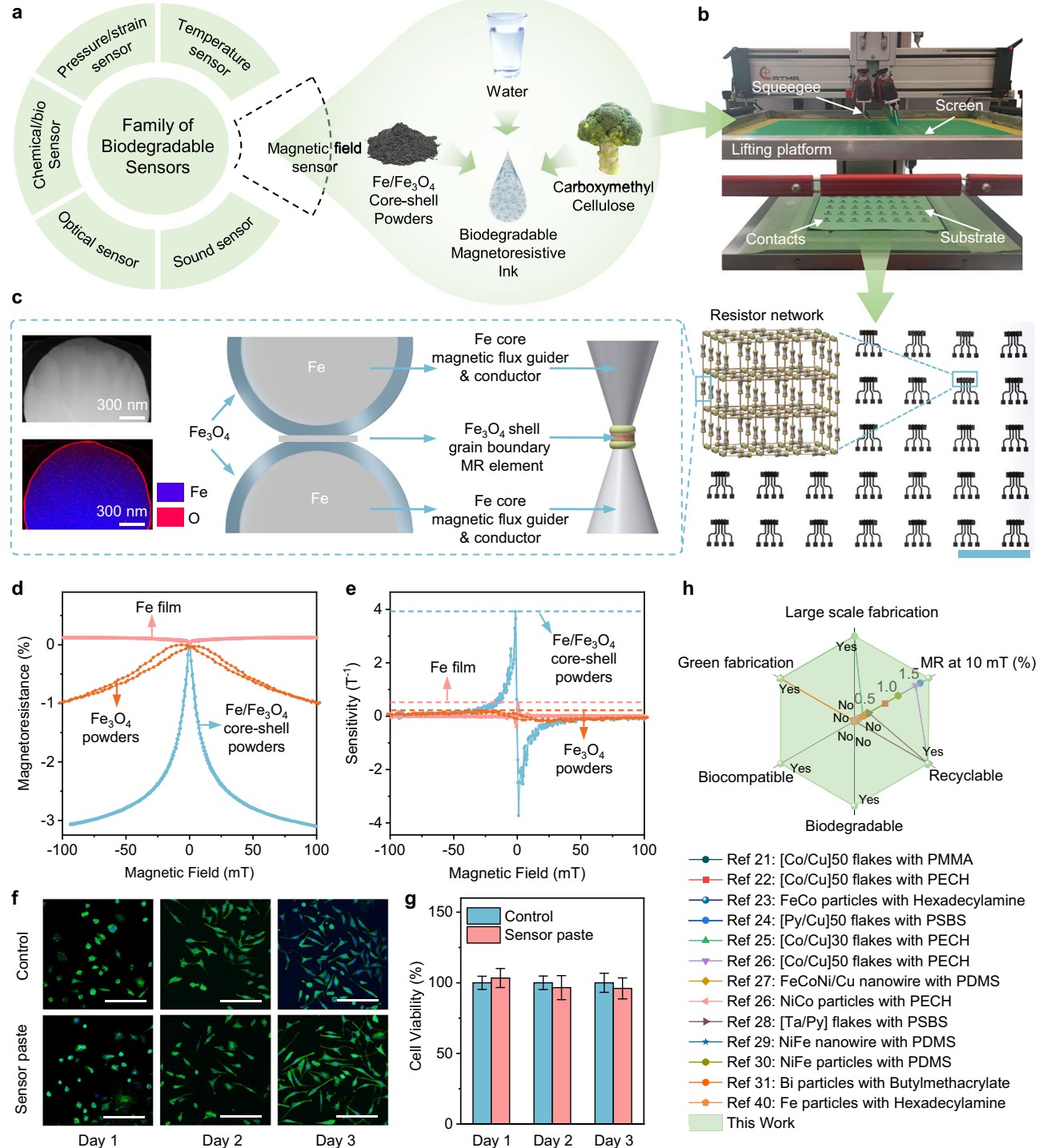

**Fig. 1 | Fully green and scalable printed biodegradable magnetoresistive (MR) sensors. a** Conceptual illustration showing the biodegradable sensors family and a recipe for the lacking MR sensor. **b**, Photograph of the fabrication instrument and the screen-printed biodegradable sensor arrays on an A4 paper substrate. Scale bar, 5 cm. Inset schematic showing the resistor network model of i-MFG enhanced GB-MR sensors in a printed matrix. **c** Schematic of the synergistic effect of i-MFG and GB-MR sensor, high-angle annular dark-field STEM image (top), and corresponding superimposed EDXS-based element distribution map (bottom) of the cross-section of an $Fe/Fe_3O_4$ core-shell microparticle. **d** Magnetoresistance and **e** sensitivity comparison of printed $Fe/Fe_3O_4$ core shell particles, printed $Fe_3O_4$ particles, and sputtered Fe thin film. **f** Fluorescence images show the normal morphology evolution of L929 cells cultured with printed sensor pieces over 3 days. Green indicates the cytoskeleton, and blue indicates the nucleus. Scale bars: 200 μm. **g** Quantitative MTT assay analysis of cell viability and comparison over 3 days. Error bars are

presented as mean ± standard deviation (SD). **h** Comparison of different printed MR sensors in terms of the large-scale and green fabrication (non-toxic solvent and energy saving), biodegradable, biocompatible, recyclable, and MR ratio at 10 mT. A quantitative comparison of the literature-reported sensors (including fabrication process, functional elements, binders, and low-field metrics) is provided in Supplementary Table 1. The corresponding functional fillers and binders in references: Ref. 21: $[Co/Cu]_{50}$ flakes with poly(methyl methacrylate) (PMMA); Ref. 22: $[Co/Cu]_{50}$ flakes with polyepichlorohydrin (PECH); Ref. 23: FeCo particles with Hexadecylamine; Ref. 24: $[Py/Cu]_{50}$ flakes with poly(styrene-butadiene-styrene) (PSBS); Ref. 25: $[Co/Cu]_{30}$ flakes with PECH; Ref. 26: $[Co/Cu]_{50}$ flakes with PECH; NiCo particles with PECH; Ref. 27: FeCoNi/Cu nanowire with polydimethylsiloxane (PDMS); Ref. 28: $[Ta/Py]$ flakes with PSBS; Ref. 29: NiFe nanowire with PDMS; Ref. 30: NiFe particles with PDMS; Ref. 31: Bi particles with Butylmethacrylate; Ref. 40: Fe particles with Hexadecylamine.

Owing to the removal of toxic chemicals from the ink formulation and the prevention of harmful by-product generation during the mild printing process, our sensor platform achieves high biocompatibility. To experimentally validate this property, standard in vitro biocompatibility tests are applied, in which printed sensor pieces are cultured with a mouse fibroblast cell line (L929 cells). Cell morphology and proliferation are recorded, and the cytoskeleton is examined by immunofluorescence staining over a 3-day period. The L929 cells exhibit normal spreading behavior, with increased cell density and a typical filamentous, stretched morphology on days 2 and 3 (Fig. 1f). No obvious differences in cell morphology, distribution, or density are observed between the $Fe/Fe_3O_4$-NaCMC group and the control group. A quantitative 3-(4,5-dimethylthiazol-2-yl)−2,5-diphenyltetrazolium bromide (MTT) assay analysis shows that cells cultured with $Fe/Fe_3O_4$-NaCMC sensor pieces retain a viability exceeding 95% throughout the 3-day period, confirming the absence of cytotoxic effects and biocompatibility of the printed sensor pieces (Fig. 1g).

Importantly, the biocompatible sensor exhibits remarkable sensing performance, featuring a magnetoresistance ratio of about −3.1 % at 100 mT, with a maximum sensitivity of 3.93 $T^{-1}$ at 1.3 mT (Fig. 1d, e). When benchmarked against Fe films prepared by high-vacuum magnetron sputtering, these sensing metrics are about 27.5 times and 8 times higher, respectively. Our sensors also outperform printed $Fe_3O_4$-microparticle-based sensors by factors of 3.4 in magnetoresistance and 20 in sensitivity, and 200 in electrical conductivity. The outstanding sensing performance and electrical conductivity stem from a synergistic strategy involving systematic material and structural engineering of $Fe/Fe_3O_4$ core-shell microparticles, enabling spin-dependent hopping-mediated $Fe_3O_4$ grain boundary magnetoresistance, in situ Fe-core magnetic flux guider (i-MFG), and highly conductive Fe connector (Fig. 1c), as fully validated in the following discussion. These designs work in concert, delivering exceptional low-field response, as evidenced by a higher magnetoresistance ratio at 10 mT relative to all printed magnetoresistive sensors reported previously[21–31,40] (Fig. 1h and Supplementary Table 1).

At the end of their service life, the sensors exhibit eco-friendly recyclability and inherent biodegradability, as demonstrated in subsequent assessments. Building on these technical advancements, the proposed sensor platform establishes a new paradigm for magnetoelectronics by concurrently addressing lifespan sustainability and high sensing performance.

## Eco-friendly recycling and controllable biodegradation for environmentally responsible disposal of end-of-service sensors

Thanks to the incorporation of water-soluble NaCMC as a polymeric binder and hosting substrate, the discarded sensors exhibit rapid disintegration within 1 h upon exposure to water. This eliminates the need for any harsh physical treatment or corrosive reagents commonly used in conventional recycling protocols[41]. The intrinsic magnetism of the released $Fe/Fe_3O_4$ microparticles allows for their efficient collection using permanent magnets, thereby promoting a closed-loop recycling process (Fig. 2a and Supplementary Movie 2). To demonstrate the versatility of this approach, we also successfully recover sensors printed on paper substrates (Fig. 2a). Notably, even when the fillers consist of multiple components (e.g., Mo powders released from the printed electrodes), these mixed fillers can be readily separated via magnetic sorting (Fig. 2a and Supplementary Movie 3). Notably, under practical recycling conditions (water exposure <1 h per cycle), the reclaimed $Fe/Fe_3O_4$ microparticles retain their functional quality, as evidenced by the essentially unchanged MR performance of reprinted sensors over multiple recycling cycles (Supplementary Fig. 4), confirming the excellent reusability of the magnetic fillers.

However, under real-world conditions, the efficient retrieval of discarded electronic waste remains a major obstacle to recycling. Addressing this concern, our sensor platform is designed for safe

disposal. Upon release into the environment, the $Fe/Fe_3O_4$ microparticles gradually degrade into Fe ions over time, without generating any hazardous by-products[42,43]. Nevertheless, the water-sensitive nature of the printed sensor is a double-edged sword because it may compromise long-term operational stability, particularly in humid or aqueous environments. To address this, various biodegradable protective layers are investigated to regulate the water-triggered degradation behavior (Fig. 2b, c). Among these, the sensor encapsulated with Ecoflex exhibits the most robust performance, retaining their magnetoresistance with minimal degradation for up to 90 days. In contrast, the beeswax-encapsulated sensor maintains a relatively stable magnetoresistance for the first 30 days, which then declines to 2.7% after 60 days, eventually leading to device failure by the 90th day. The sensor sealed with cellulose acetate shows the shortest lifetime (about 3 days). Consistent with these immersion results, accelerated aging under controlled humidity and temperature further confirms that encapsulation markedly expands the operational window from ambient conditions to high-humidity and warm-water environments (Supplementary Fig. 5). These results demonstrate that encapsulation strategies can be rationally tailored to customize sensor degradation kinetics and functional lifespan, thereby meeting different requirements across diverse applications.

The sensor platform is further validated by incorporating alternative biocompatible and even edible binders (e.g., sodium alginate, egg white, starch). These materials not only ensure biosafety but also provide tunable degradation profiles. Sensors fabricated with these binders exhibit stable and reliable performance, with magnetoresistance ratios maintained around −3% and sensitivities exceeding 3.5 $T^{-1}$ (Fig. 2d, Supplementary Fig. 6). The combination of green fabrication, biocompatibility, and biodegradability broadens the scope of printable substrates beyond traditional polyethylene terephthalate (PET) foils, paper and ceramics (Supplementary Fig. 7). Our sensors can even be directly printed on sensitive biological objects with various surface textures such as tomato, leaf and daisy petals (Fig. 2e, Supplementary Fig. 8). Despite the mechanical and chemical variability of these natural substrates, the printed sensors exhibit consistent performance with magnetoresistance above −2.9 % and sensitivity exceeding 3.2 $T^{-1}$. After service, these sensors can be completely removed by simple rinsing with water, without leaving residue or damage to their surfaces (Fig. 2e). The controllable biodegradation, coupled with the eco-friendly recycling, will ensure a more environmentally responsible end-of-life management for the sensor.

## Spin-dependent hopping-mediated grain boundary magnetoresistance in printed $Fe/Fe_3O_4$ core-shell microparticle matrix

To overcome the intrinsically low magnetoresistance of metallic Fe-based materials, $Fe_3O_4$ grain boundary magnetoresistance is engineered to enhance sensing performance. However, the inherently high resistivity of $Fe_3O_4$ poses a challenge for printable electronics, which typically suffer from limited resolution and require sufficient material conductivity to build electrical percolation. To address this conflict, we designed the $Fe/Fe_3O_4$ core-shell microparticles. In this architecture, the metallic Fe core provides a highly conductive backbone that drastically reduces the overall resistance of the composite, while the $Fe_3O_4$ shell accommodates magnetoresistive response through spin-dependent transport across grain boundaries. Compared with composites printed with $Fe_3O_4$ microparticles, the $Fe/Fe_3O_4$ sensors exhibit resistance reduction by over two orders of magnitude (Supplementary Table 3), rendering them suitable for practical electronic applications.

Intriguingly, the printed $Fe/Fe_3O_4$ sensor demonstrates nonlinear current-voltage characteristics (Fig. 3a, Supplementary Fig. 9), consistent with a variable-range hopping transport mechanism involving spin-dependent interfacial barriers[23,44–46]. Following the Efros-Shklovskii variable-range hopping model[23,44–46], the application of an increasing electric field (via higher bias voltage) reduces the energy

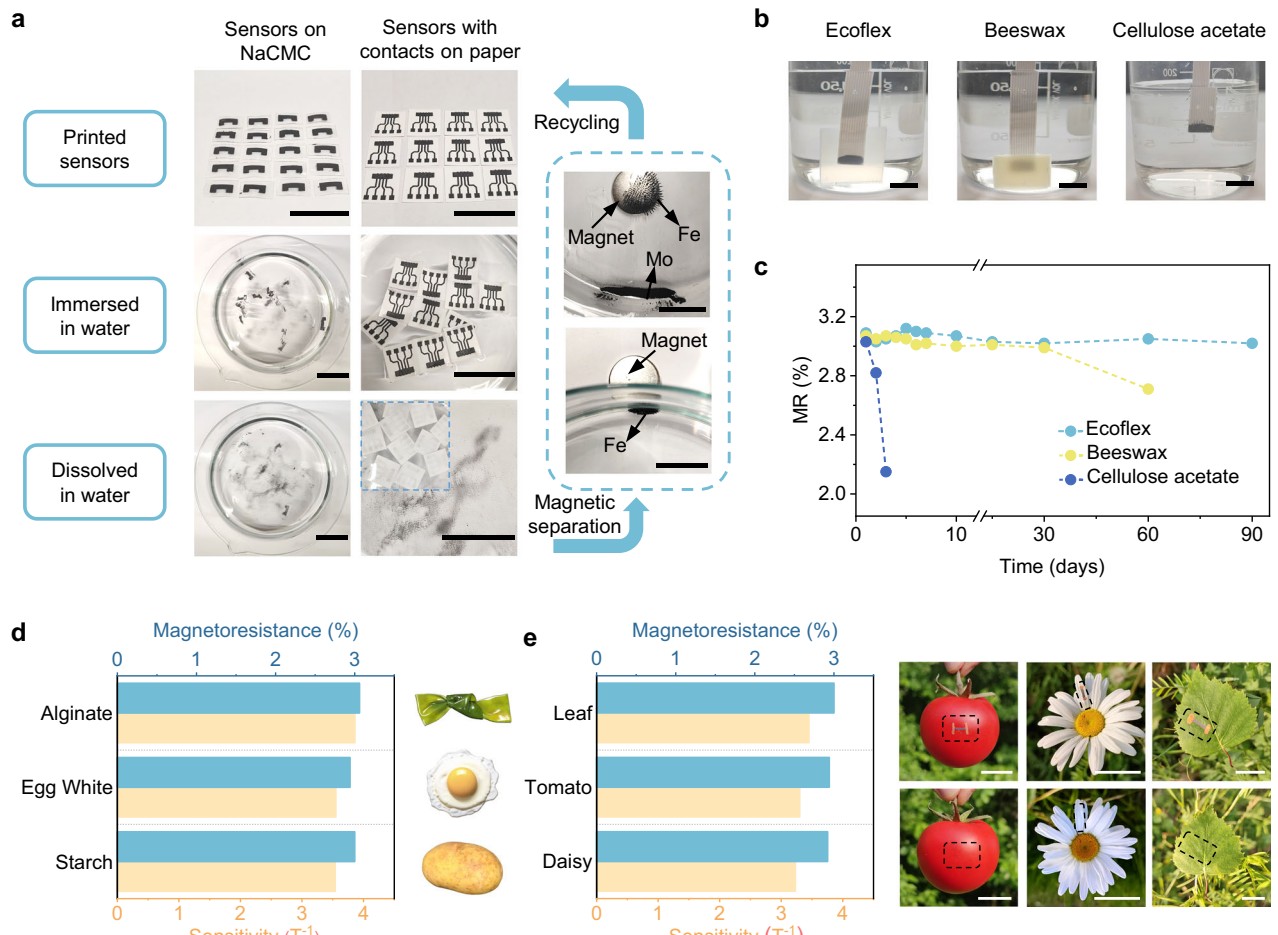

**Fig. 2 | Eco-friendly recycling and biodegradability of the printed Fe/Fe$_3$O$_4$-NaCMC MR sensors. a** Photographs of printed Fe/Fe$_3$O$_4$-NaCMC sensors on NaCMC and paper substrates with Mo contacts, demonstrating dissolution in water at room temperature. Photographs in dashed block showing the separation and recovery of dispersed Fe/Fe$_3$O$_4$ microparticles using a permanent magnet. Scale bar, 3 cm. **b** Photographs showing printed Fe/Fe$_3$O$_4$-NaCMC sensors with different biodegradable encapsulation materials immersed in water. Scale bar, 1 cm. **c** MR performance degradation of printed Fe/Fe$_3$O$_4$-NaCMC sensors with different biodegradable encapsulation materials during water immersion at room temperature. **d**, Magnetoresistance and sensitivity of printed Fe/Fe$_3$O$_4$ sensors with different binders: alginate, egg white, and starch. **e** Magnetoresistance and sensitivity of printed Fe/Fe$_3$O$_4$-NaCMC sensors on different biological objects substrates (leaf, tomato, and daisy petal). The bottom photographs show the comparison after washing off the printed sensors placed on the different biological objects substrates. Scale bar, 2 cm.

barrier associated with electron hopping across grain boundaries (Supplementary Fig. 10). This facilitates long-range electron hopping and results in a nonlinear increase in current. In stark contrast, selective removal of the Fe$_3$O$_4$ shell using acidic treatment leads to a transition to linear current-voltage behavior (Supplementary Fig. 9), indicative of ohmic conduction dominated by the Fe core. Further support for the variable-range hopping mechanism is provided by temperature-dependent resistance measurements, where the resistance ($R$) scales with temperature ($T$) (Fig. 3b, Supplementary Fig. 11), following $\ln(R) \propto T^{-1/2}$ (Fig. 3c, Supplementary Fig. 11). This relationship aligns well with the Efros−Shklovskii variable-range hopping transport model[23,44–46], in which the electron hopping between localized states is controlled by the barrier height. The magnetoresistive behavior of the Fe/Fe$_3$O$_4$ core-shell particle sensors can be attributed to spin-dependent hopping across magnetically disordered interfaces between adjacent Fe$_3$O$_4$ shells. Upon application of an external magnetic field, the alignment of local magnetic moments reduces spin disorder and enhances carrier hopping mobility, giving rise to negative magnetoresistance effects[47,48]. In particular, as the applied bias increases from 0.1 V to 5 V, the magnetoresistance is progressively suppressed (Fig. 3d, Supplementary Fig. 10), which is attributed to the reduction in the effective spin-dependent hopping barrier under

stronger electric fields. As a result, the enhanced hopping of minority-spin carriers lowers the overall spin polarization of the transport electrons, resulting in a gradual suppression of the magnetoresistance ratio.

To further enhance the grain boundary magnetoresistance at Fe$_3$O$_4$ shell interfaces, a controlled thermal oxidation strategy is employed to optimize the structural and magnetic quality of the oxide shell (Fig. 3e). Specifically, a moderate vacuum level (0.5 mbar) is purposely selected to finely regulate the oxidation kinetics, ensuring the formation of nanoscale high-quality Fe$_3$O$_4$ shells. The balanced oxygen supply avoids overly rapid oxidation that could lead to undesirable phase formation or excessive structural disorder. Subsequent tuning of the oxidation temperature enables precise control over shell thickness and crystallinity, both of which are crucial for modulating spin-dependent transport across grain boundaries. As a reference, Fe microparticles oxidized at ambient conditions possess a thin Fe$_3$O$_4$ layer (<6 nm), accompanied by a high density of antiphase boundaries[49] (Fig. 3f, Supplementary Fig. 12). These antiphase boundaries act as spin-flip scattering centers that severely limit spin polarization[37,49–51], thus suppressing the magnetoresistance ratio to only −0.92% in the corresponding sensors. Increasing the oxidation temperature promotes the growth of a coherent and thicker Fe$_3$O$_4$

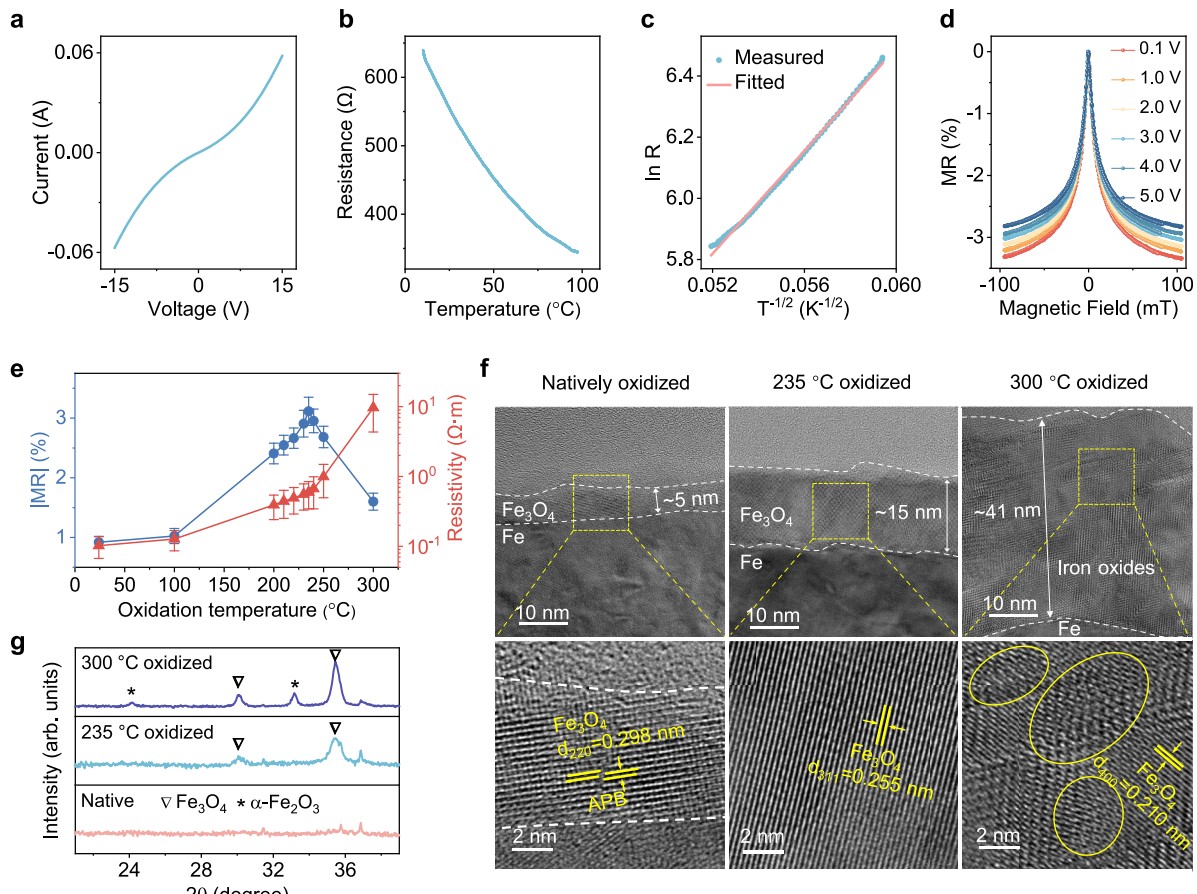

**Fig. 3 | Spin-dependent hopping-mediated grain boundary magnetoresistance. a** I-V characteristics of printed Fe/Fe$_3$O$_4$-NaCMC sensors with Fe particles thermally oxidized at 235 °C under medium vacuum. **b** Variation in resistance with temperature of printed Fe/Fe$_3$O$_4$-NaCMC sensors with Fe particles thermally oxidized at 235 °C under medium vacuum. **c** Logarithm of the electrical resistivity as a function of $T^{-1/2}$ of the curve in (**b**). **d** Voltage-dependent magnetoresistance of printed Fe/Fe$_3$O$_4$-NaCMC sensors by Fe particles thermally oxidized at 235 °C under medium vacuum. **e** MR performance and electrical resistivity of printed

Fe/Fe$_3$O$_4$-NaCMC sensors fabricated with the Fe particles thermally oxidized at different temperatures under medium vacuum. Error bars are presented as mean ± SD. **f** HR-TEM images of printed MR sensors fabricated with Fe particles thermally oxidized at different temperatures under medium vacuum. **g** XRD characteristics of Fe particles thermally oxidized at different temperatures under medium vacuum. Diffraction angle from 20° to 40° where the characteristic reflections of iron oxide occur. The full diffractograms are shown in Supplementary Fig. 13.

shell (10–20 nm), reflected by a remarkable increase in electrical resistivity (Fig. 3e). High-resolution transmission electron microscopy (HR-TEM) reveals well-defined lattice fringes and a notable reduction in antiphase boundary density within these thermally oxidized shells (Fig. 3f). These structural improvements result in enhanced spin coherence and interfacial spin filtering, leading to an improved magnetoresistance ratio of −3.1% at an optimal oxidation temperature of 235 °C. However, excessive oxidation (e.g., at 300 °C) induces the formation of insulating antiferromagnetic α-Fe$_2$O$_3$ phase[52], as confirmed by X-ray diffraction (XRD) and Raman spectroscopy (Fig. 3g, Supplementary Figs. 13, 14), which disrupts spin-polarized transport (Fig. 3e). Moreover, the accelerated oxidation process tends to introduce additional structural defects (Fig. 3f), eventually reducing the magnetoresistance ratio to about −1.6% (Fig. 3e). These findings highlight the delicate interplay between oxidation environment, shell composition and microstructure in dictating the spin-dependent hopping behavior, offering a tunable pathway to further optimize the magnetoresistive performance.

**In situ magnetic flux guiding effect to boost low-field sensitivity**
Although engineered Fe$_3$O$_4$ grain boundaries enable significant spin-dependent transport and exhibit a substantial magnetoresistance

ratio, this grain boundary–induced magnetoresistance effect of Fe$_3$O$_4$ typically requires the application of strong magnetic fields (>1 T) to be effective[34–37] (Supplementary Table 3). To address this limitation, a structural core-shell configuration was designed for the magnetoresistive fillers within the composite. By leveraging the intrinsically high magnetic permeability of metallic Fe, the Fe core encapsulated in Fe$_3$O$_4$ shell is purposely designed to function as an in situ magnetic flux guider, which locally amplifies the magnetic flux density (that is, characterized by a gain factor) and thus enhances the sensitivity of the Fe/Fe$_3$O$_4$ sensor at weak external magnetic fields. This hypothesis is strongly supported by the experimental results (Fig. 4a, b and Supplementary Fig. 15), for example, as numerically dividing a gain factor of 10 from the magnetoresistance response of the Fe/Fe$_3$O$_4$ sensor, the resulting curve closely resembles that of the Fe$_3$O$_4$ sensor. To elucidate the underlying mechanism, micromagnetic simulations are conducted. As illustrated in Fig. 4c, the Fe cores are capable of concentrating magnetic flux lines, thus altering the spatial distribution of magnetic flux density, especially at Fe$_3$O$_4$ grain boundaries. Considering the wide size distribution of the Fe/Fe$_3$O$_4$ microparticles, a representative geometry (featuring a 240-nm-diameter Fe core and a 10-nm-thick Fe$_3$O$_4$ shell) is selected for simulation. In this case, the local gain factor at the Fe$_3$O$_4$ region can reach 14 (Fig. 4c). Additional

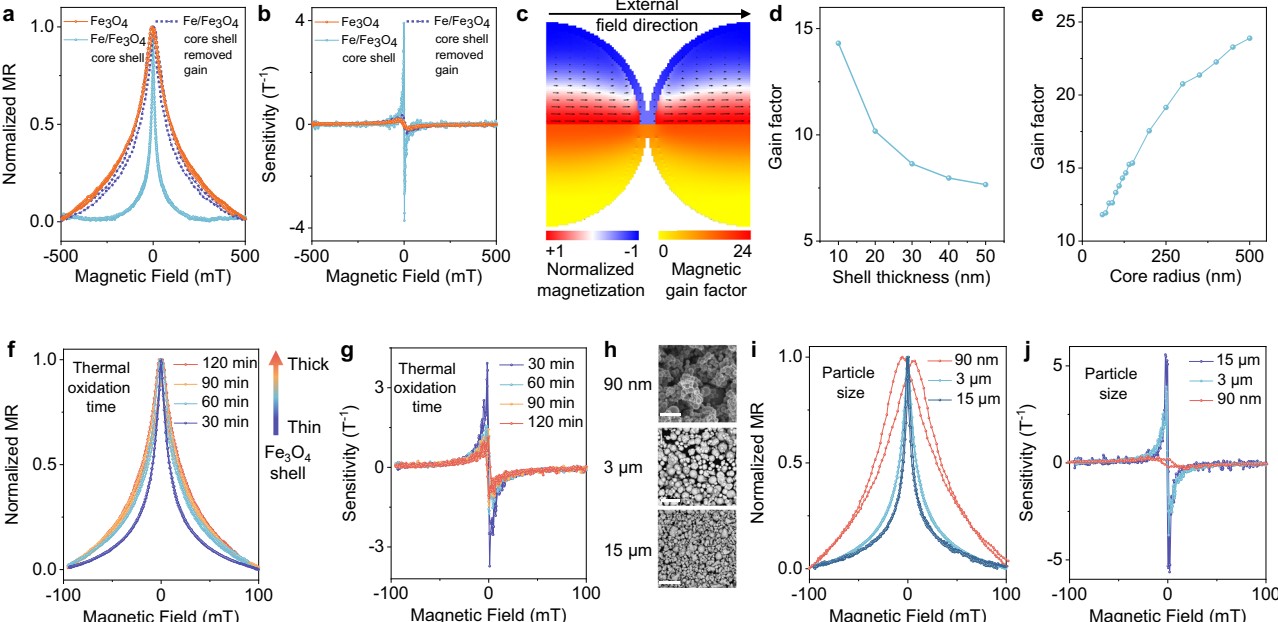

**Fig. 4 | In situ magnetic flux guiding effect to boost low-field sensitivity.**
**a** Normalized MR curves and **b** sensitivity curves comparison of printed Fe/Fe₃O₄ core-shell particles (cyan curve), with which removed 10 gain factor (purple dashed line curve) and printed Fe₃O₄ particles (red curve). **c** Magnetic state and magnetic gain factor at the junction of two spheres with the core radius 120 nm and shell thickness 10 nm (simulations, cubic mesh of 2.5 nm size). A magnetic field of 100 mT is applied along the symmetry axis of the system. The top part shows the equilibrium magnetic state with the color-code indicating $M_x$ component of magnetization normalized by the saturation magnetization of the Iron core and arrows indicating magnetization direction. The bottom part shows the magnetic gain factor calculated based on the $B_x$ component of the local value of the stray field related to the external field. **d** Gain factor as a function of the shell thicknesses of the core-shell spheres with the fixed core diameter of 240 nm and magnetic field

100 mT applied along the x-axis. The gain factor is calculated for the average B-field at the junction between spheres. **e** Same, as a function of the core radius of the core-shell spheres with the fixed shell thickness of 10 nm. **f** Normalized MR curves and **g** sensitivity curves of printed Fe/Fe₃O₄ core shell particles with different thicknesses that were thermally oxidized for different times (30 min, 60 min, 90 min, and 120 min along the arrow) at 235 °C. **h** SEM images of Fe/Fe₃O₄ core-shell particles with different sizes (90 nm, 3 μm, and 15 μm), Scale bars: 500 nm, 10 μm, 100 μm from top to bottom. For the distribution of particle diameters, see Supplementary Figs. 19 and 20. All three size classes were oxidized under the same conditions (235 °C, medium vacuum 0.5 mbar, 30 min) to keep the oxide-shell formation comparable across samples. **i** Normalized MR curves and **j** sensitivity curves of printed Fe/Fe₃O₄ core-shell particles with different sizes.

simulations further indicate that a reasonable size mismatch between two contacting particles introduces only a modest variation in the local gain factor (Supplementary Fig. 16), whereas the gain factor is highly sensitive to the relative orientation between the external magnetic field and the percolation path connecting adjacent particles (Supplementary Fig. 17 and 18). As the alignment shifts from parallel to tilted, the gain factor decreases progressively. Due to the random arrangement of particles within the composite, the overall averaged effect is expected to be lower, which is in accordance with the experimentally observed gain factor of approximately 10.

To shed light on future device optimization, a comprehensive investigation combining numerical simulations and experimental studies is conducted (Fig. 4d–j). Firstly, the dependence of magnetic flux density gain on the Fe₃O₄ shell thickness is theoretically evaluated. For a fixed Fe core diameter, the gain factor is found to decrease gradually with increasing shell thickness (Fig. 4d). That is because the increased shell volume introduces a larger region of comparatively lower magnetic permeability, which disperses the magnetic flux lines and diminishes the localized field enhancement effect. To experimentally validate this trend and correlate it with magnetoresistance behavior, shell thicknesses are systematically modulated by varying the thermal oxidation time while maintaining constant temperature and vacuum conditions. The resulting magnetoresistance curves exhibit pronounced broadening and sensitivity decreasing with increasing shell thickness (Fig. 4f, g and Supplementary Fig. 15), indicative of a reduced flux-concentrating effect. Subsequently, the effect of core size on magnetic flux enhancement is investigated under the condition of a constant Fe₃O₄ shell thickness. The simulation results indicate that

larger Fe cores lead to an increase in the gain factor, which eventually approaches a saturation value (Fig. 4e). That is because as the core diameter increases, the greater permeability contrast improves flux concentration. However, beyond a critical size, the resulting flux distribution reaches a geometric limit, no longer enhancing the field in grain boundaries. This simulated trend is confirmed by experimental observations. As shown in Fig. 4h, i and Supplementary Fig. 15, composites incorporating particles with larger core sizes exhibit noticeably sharpened (from 90 nm to 3 μm particle diameter) and then saturated (from 3 μm to 15 μm diameter) magnetoresistance responses (Fig. 4i), also reflected in sensitivity (Fig. 4j). Therefore, achieving optimal magnetic flux guidance (and hence maximum sensitivity) requires a finely tuned core-shell architecture, wherein the shell is sufficiently thin to enable strong interfacial flux transfer while the core should be relatively large to enable substantial flux concentration.

## Application in disposable magnetoelectronics

The biocompatible materials-based ink, combined with the fully green printing fabrication, the biocompatible application potential, and the inherent biodegradability, presents a viable pathway toward closed-loop sustainability for our sensors (Fig. 5a). This integrated strategy facilitates the large-scale deployment of magnetoresistive sensors with a minimized environmental footprint.

In the IoT context, smart packaging enabled by disposable magnetoresistive sensors represents a critical application area. As a proof of concept, a printed sensor, paired with a printed sustainable magnet, is affixed to a medicine box to monitor the opening frequency (Fig. 5b, c). This is particularly valuable for tracking medication adherence of

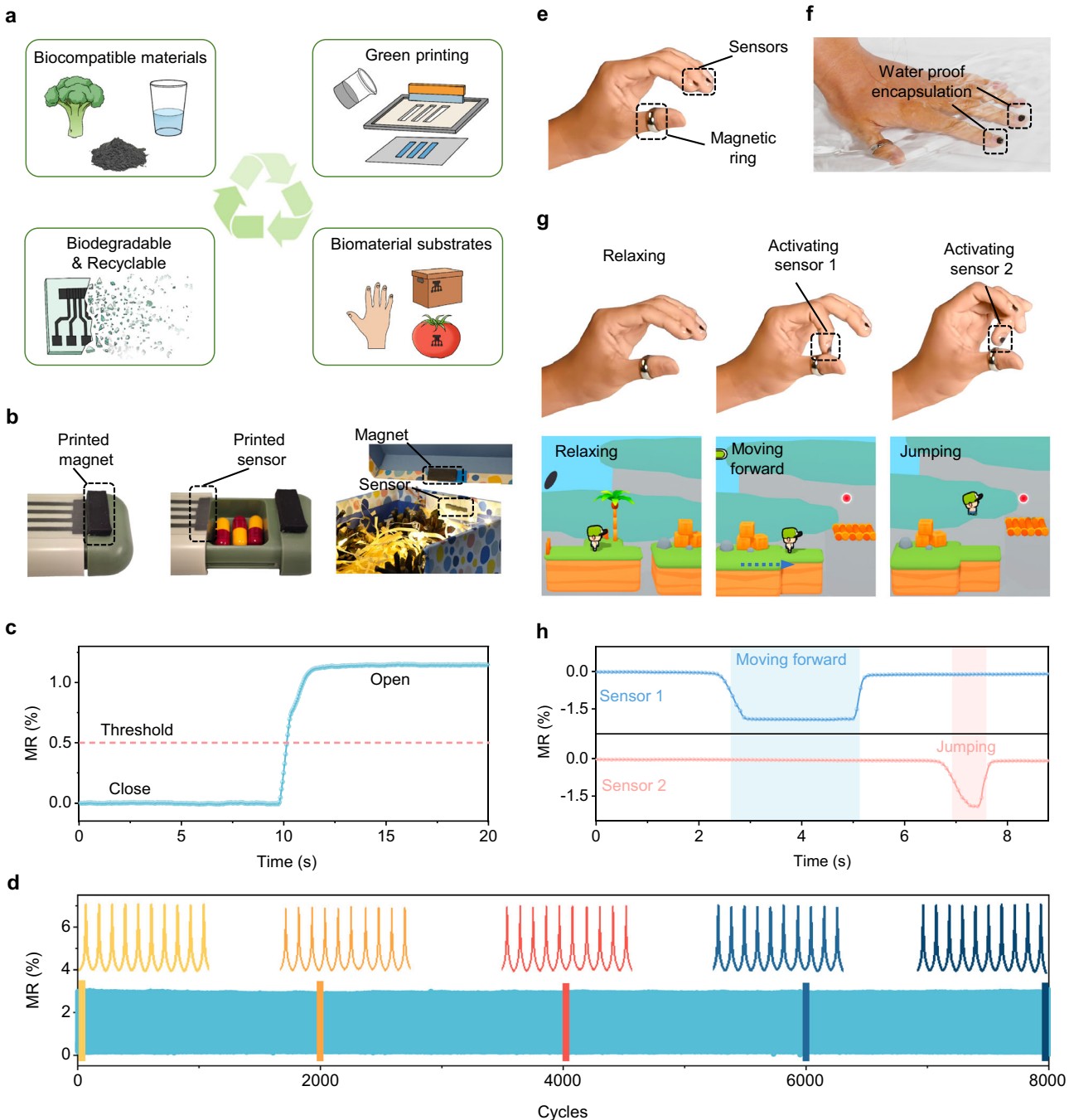

**Fig. 5 | Application in disposable magnetoelectronics. a** Conceptual illustration showing a pathway toward closed-loop eco-sustainability of the biodegradable MR sensors. **b** Smart package of a medicine box and a gift box. The left two photographs show the closed and open states of the medicine box mounted with a printed Fe/Fe$_3$O$_4$-NaCMC sensor and a printed sustainable magnet. Once the medicine box is opened, the state can be recorded by the MR sensor. The right photograph shows a gift box mounted with a printed Fe/Fe$_3$O$_4$-NaCMC sensor and a printed sustainable magnet. When the box is opened, the state can be read by the MR sensor, and the circuit control LED light is on. **c** Magnetoresistance signal variation as the box is closed and open states. **d** Operational stability test for the printed magnetoresistive sensor under a 100 mT alternating magnetic field cycles. MR performance shows no degradation after 8000 alternating magnetic field cycles. **e**–**h** Disposable human machine interface (HMI) application. **e**, Photograph of the wearable disposable MR sensor printed on the fingernail. A permanent magnetic ring worn on the thumb provides a magnetic field. **f** A gelatin-glycerol gel encapsulating the printed sensor to protect it from exposure to water. **g** Photographs of different gestures and the corresponding game operation screenshot, and **h** the time evolution of the MR signals of the two sensors read out dependent on the distance between the finger nail and the magnetic ring.

elderly individuals with memory decline, as real-time monitoring of the medicine box via an IoT system provides caregivers and healthcare providers with critical data to enable timely intervention. In a similar operational fashion, the sensor can be incorporated into a circuit involving light-emitting diodes, enabling automatic illumination triggered by the opening of a gift box (Fig. 5b). At the end of its service, the entire package can be discarded as biodegradable waste with minimal environmental impact. For practical use, the sensors exhibit reliable operational stability, retaining consistent magnetoresistance after 8000 magnetic field cycles at 100 mT (Fig. 5d). These properties unlock new opportunities for the development of next-generation disposable magnetoelectronic systems.

The validated biocompatibility of the sensor allows for direct printing onto biological objects, thereby expanding their adaptability to interactive interfaces for wearable and implantable electronics. As a representative example, an electronic interface, comprising two sensors printed on fingernails and a magnetic ring, is used for human-computer interaction (Fig. 5e–h). To ensure durability under daily usage conditions, the sensors are encapsulated in biocompatible gelatin glycerol gel, allowing stable operation while remaining removable through routine handwashing (Fig. 5f and Supplementary Movie 4). By altering the relative sensor-magnet distance, the electrical signals change accordingly (Fig. 5g, h). Once surpassing a predefined threshold, specific actions are initiated for the digital character such as move forward and jump (Fig. 5g, h and Supplementary Movie 4).

## Discussion

In summary, this work presents a comprehensive strategy that seamlessly integrates environmental sustainability with superior sensing performance across the entire lifecycle of printable magnetoresistive sensors. By employing industry-compatible screen-printing techniques and eliminating the use of harsh treatment and harmful chemicals, the fabrication process is both scalable and eco-friendly. The sensors, composed solely of naturally derived materials, inherently provide biocompatibility, biodegradability, and environmentally benign recyclability, thereby expanding their applicability to safety-critical scenarios such as food monitoring, plant/human wearables or implantables, and direct on-skin electronics. The synergistic combination of spin-dependent hopping magnetoresistance at $Fe_3O_4$ shell-to-shell grain boundaries and Fe-core-induced magnetic flux concentration and conductivity enhancement results in significantly improved low-field sensitivity, enabling practical and reliable sensor functionality. Leveraging these technical achievements, this research represents an important advancement in magnetoresistive sensor technology. Future work would benefit from a comprehensive quantitative life cycle assessment (LCA) to further quantify cradle-to-grave impacts and guide continued optimization. Taken together, this work paves the way for disposable magnetoelectronics and sustainable IoT applications.

## Methods

### Materials

The Fe particles (10 μm, Sigma Aldrich; EMSURE, average 3 μm as confirmed by scanning electron microscopy (SEM) imaging, Supplementary Fig. 20) were thermally oxidized at 235 °C under a medium vacuum (0.5 mbar) for 30 min to fabricate $Fe/Fe_3O_4$ core-shell microparticles (Supplementary Figs. 21 and 22). Without particular notice, this oxidation condition was used. Various oxidation temperatures, durations and conditions were explored to optimize magnetoresistive (MR) performance (Fig. 3e, Supplementary Figs. 23 and 24). Fe particles (<60 μm, Sigma Aldrich; average 15 μm as confirmed by SEM imaging) and Fe nanoparticles (Sigma Aldrich; average 90 nm as confirmed by SEM imaging) were thermally oxidized under the same conditions as the 3-μm particles mentioned above. Sodium carboxymethyl cellulose (NaCMC) (Mw 90,000, Sigma Aldrich) and alginic acid sodium salt (Sigma Aldrich) were dissolved in hot (60 °C) water at a weight ratio of 10% to obtain solutions. Starch (soluble, from potato, Sigma Aldrich) was dissolved in boiling water for 2 h at a 10% weight ratio to obtain a solution. Egg white was obtained from food eggs, with a typical water content of approximately 88%. Vitamin C (L-ascorbic acid, Sigma Aldrich) was dissolved in water at varying concentrations. Cellulose acetate (Mn ~50,000, Sigma Aldrich) was dissolved in acetone at a 30% weight ratio to obtain a solution (RT). Glycerol, beeswax, gelatin from porcine skin (gel strength 300, Type A), $Fe_3O_4$ powders (<5 μm), strontium ferrite powders, molybdenum powder (1–5 μm) from Sigma Aldrich, and Ecoflex 00-35 (parts A and B) were used as supplied. Fe thin films were deposited via magnetron sputtering, with a

thickness of approximately 1.5 μm, as confirmed by confocal laser scanning microscopy (CLSM, Zeiss).

### Fabrication of biodegradable magnetoresistive inks and sensors

$Fe/Fe_3O_4$ core-shell microparticles were uniformly dispersed in a NaCMC aqueous solution at a 60% volume ratio (Fe in the final composite) to form the $Fe/Fe_3O_4$-NaCMC ink. Inks with various volume ratios from 40% to 80% and alternative binders (egg white, starch, and alginic acid sodium salt) were also prepared. Without particular notice, 60% volume ratio and NaCMC binder were used. A flat flexible cable (FFC) served as the standard substrate/electrode for testing throughout. Ink was deposited using a 100 μL pipette (VWR) and then air-dried without further treatment. The FFC featured electrode slots spaced 0.5 mm apart, with a width of 5 mm and the printed sensor piece thickness of approximately 100 μm (Supplementary Fig. 20). In addition to FFCs, inks were printed on conventional substrates (plastic, paper, ceramics) and biological-substrates (tomato, leaf, daisy petals). For large-scale printing, a commercial automatic screen-printing instrument (Kundisch GmbH & Co. KG) was used to print 5×8 sensor arrays on A4-sized paper sheets.

### Rheology characterization

Rheological measurements were performed to evaluate the viscoelastic behavior of the inks. The MCR 302 rheometer (Anton Paar) equipped with the MRD180/1 T magnetocell in a plate–plate configuration (1 mm gap) was used to determine the shear-thinning and oscillatory behavior of the inks. All measurements were performed at 22 °C and the sample volume was constant.

### Vitamin C treatment for oxide shell removal

A biocompatible acidic post-treatment was employed to remove the oxide shell of $Fe/Fe_3O_4$ core-shell particles in the printed composite, enabling investigation of its role in MR performance. A biocompatible reducing acid, Vitamin C (L-Ascorbic Acid), was dissolved in water at varying concentrations. The acid solution was then drop-cast onto the dried $Fe/Fe_3O_4$-NaCMC pattern. Post-treated samples were analyzed after drying. The MR performance of the post-treated samples with different Vitamin C concentrations is shown in Supplementary Fig. 25.

### Materials characterization

**XRD**. X-ray diffraction studies were carried out using a Rigaku SmartLab 3 kW in Bragg-Brentano geometry with Cu-Kα radiation (average wavelength: 1.542 Å) and a Ni filter. The HyPix detector was employed in 1D TDI mode with XRF reduction enabled. The PDF standard cards of α-Fe (#00-006-0696), $Fe_3O_4$ (#01-084-6684), and α-$Fe_2O_3$ (#00-001 1053) were used to index the reflections.

**TEM**. To characterize the microstructure of the as-prepared and thermally oxidized samples, transmission electron microscopy (TEM) was performed. To this end, cross-sectional TEM lamella preparation was carried out by in situ lift-out using a Helios 5 CX focused ion beam (FIB) device (Thermo Fisher). To protect the sample surface, a carbon cap layer was deposited beginning with electron-beam-assisted and subsequently followed by Ga-FIB-assisted precursor decomposition. Afterwards, the TEM lamella was prepared using a 30-keV Ga-FIB with adapted currents. Its transfer to a 3-post copper lift-out grid (Omniprobe) was done with an EasyLift EX nanomanipulator (Thermo Fisher). To minimize sidewall damage, Ga ions with only 5-keV energy were used for final thinning of the TEM lamella to electron transparency. Bright-field and high-resolution TEM (HR-TEM) imaging were done using an image-$C_s$-corrected Titan 80-300 microscope (FEI) operated at an accelerating voltage of 300 kV. High-angle annular dark-field scanning transmission electron microscopy (HAADF-STEM) imaging and spectrum imaging analysis based on energy-dispersive X-ray spectroscopy (EDXS) were performed at 200 kV using a Talos F200X

microscope equipped with a Super-X EDX detector system (FEI). Prior to (S)TEM analysis, each specimen mounted in a high-visibility low-background holder was placed for 8 s into a Model 1020 Plasma Cleaner (Fischione) to remove potential contaminations. Determination of the formed crystalline iron oxide phases was done by fast Fourier transform (FFT) analysis of selected cross-sectional iron oxide shell regions. In particular, potential zone axis diffractograms were calculated from the recorded HR-TEM images and compared with simulated diffraction patterns of cubic magnetite[53] using the JEMS software package.

**SEM.** The microstructure and morphology of raw Fe powders and printed sensor pieces were observed using the backscattered electron detector of the Phenom XL Desktop SEM (Thermo Fisher Scientific, United States) with an acceleration voltage of 10 kV and a pressure of 0.1 Pa.

Raman spectroscopy: Raman signal was probed (HORIBA, Lab-RAM HR Evolution) using a 532 nm green laser. The incident laser is focused using a 100× objective lens with a diameter of around 1 μm. To avoid in situ oxidation of the iron powder during the measurement, the laser power is limited at 3.2 mW.

### Electrical characterization

The I-V curves were recorded using a Precision Source/Measure Unit (Keysight B2900A). The temperature-dependent resistance was measured using a Tensormeter (HZDR Innovation, Germany).

### Magnetoresistance measurement

The magnetoresistance ratio is defined as the magnetic field-dependent variation of the sensor's resistance, $R(H)$, normalized to the resistance value at zero magnetic field, $R_O$. The expression is $MR(H) = [R(H) - R_O]/R_O \cdot 100\%$. The sensitivity of the sensor's response to the magnetic field is defined as the ratio of the first derivative of the sensor's resistance with respect to the magnetic field to the resistance value, $S(H) = [dR(H)/dH]/R(H)$. The figure of merit (FoM)[24,30] is defined as the ratio of the maximum sensitivity to the magnetic field at which the maximum sensitivity occurs, $FoM = S_{max}/H_{s'max}$. A standard Helmholtz coil was used to generate a uniform and tunable magnetic field, and the applied magnetic fields were swept between ±100 mT. A four-probe configuration was used for the MR measurement of the printed sensors using a Tensormeter (HZDR Innovation GmbH, Germany). The drive current had a frequency of 775 Hz and an amplitude of 100 μA, with the magnetic field aligned parallel to the bias current of the sensor. To assess performance reproducibility, sensors from three independently fabricated batches were measured (batch 1-3, $n = 10$ devices per batch; total $n = 30$), prepared under identical conditions. All measurements were performed at room temperature.

### Magnetic simulations

We perform micromagnetic simulations of ferromagnetic nano- and microspheres using OOMMF[54] and Ubermag[55] softwares. The geometry under consideration consists of two spheres placed along the x-axis and touching each other according to the finite difference cubic mesh. The core radius $R_C$ and the shell thickness $h$ are varied. We use the saturation magnetization $\mu_0 M_{S\ core} = 2.2$ T and $\mu_0 M_{S\ shell} = 0.5$ T for the core and shell, respectively. The exchange stiffness is set to 21 pJ everywhere. The sample is exposed to an external magnetic field with a strength of $B_{out} = 100$ mT. In each simulation, the initial magnetic state is set along the field direction, then the energy minimization procedure is performed, and the stray field is calculated.

To analyze the amplification factor as a function of core radius, we considered core radii from $R_C = 30$ nm to $R_i = 500$ nm with a fixed shell thickness of 10 nm, and an external field is applied along the symmetry axis (x). In simulations with $R_i < 70$ nm, the cubic mesh is of 2.5 nm size. In simulations with radii $R_i > 300$ nm, the mesh size is 10 nm. For all

other simulations, we use a mesh of 5 nm. In the case of the very small core radii below 55 nm, we found that the equilibrium state of each sphere corresponds to the 3D onion state (the magnetization has no azimuthal component with respect to the symmetry axis of the system), while for the larger spheres the equilibrium state is the whirligig one with the vortex-like curling of magnetization around the symmetry axis[56]. In both cases, the 3D onion and whirligig states are modified by the external field. In the following, we consider the samples with $R_i > 50$ nm only. The gain factor is defined as,

$$g = \mu_0 (H_{ms} + M_{S\ shell})/B_{out}, \qquad (1)$$

where $H_{ms}$ is the mean value of the magnetic field at the interface between the spheres. Due to symmetry, the single x-component of $H_{ms}$ is nonvanishing. The amplification factor (gain) $g$ increases with $R_i$ and tends to saturate for larger radii. We repeat the same procedure to analyse the dependency of $g$ on the shell thickness $h$ fixing $R_i = 120$ nm. With the growth of $h$ from 10 nm to 50 nm, $g$ decreases by about 40%. And finally, we calculate how g depends on the direction of the external magnetic field rotated in the x-z plane. For the fixed $R_i = 120$ nm and $h = 10$ nm, we perform separate energy minimization procedures for the field angles changed from 0° (along the x-axis) to 90° (along the z- axis) with a step of 15°. With the rotation of the field, the symmetry of the state reduces, giving rise to a z-component of $H_{ms}$ with a sequential reduction of its x-component (Supplementary Fig. 17). For the field direction of 90°, there is also a y-component of the stray field at the junction point between the two spheres.

### Biodegradability characterization

The Fe/Fe$_3$O$_4$-NaCMC sensors printed on NaCMC substrate or on paper substrate with printed Mo contacts were immersed in water at room temperature. The dispersed Fe/Fe$_3$O$_4$ core-shell particles were collected and recovered by a NdFeB permanent magnet.

### Recyclability and water-immersion aging tests of reclaimed Fe/Fe$_3$O$_4$ core-shell microparticles

1. Recycling–reprinting cycle (practical recycling condition). Printed sensors were recycled by exploiting the water solubility of the NaCMC binder. Discarded sensors were immersed in deionized (DI) water until the printed layer disintegrated and the composite dispersed. The released Fe/Fe$_3$O$_4$ microparticles were collected from the suspension using a permanent magnet, rinsed with DI water to remove residual NaCMC, and dried under ambient conditions (or under mild vacuum) until constant mass. The reclaimed particles were then re-mixed in a freshly prepared NaCMC aqueous solution to formulate inks following the same protocol as for the pristine ink. The ink was subsequently reprinted to fabricate new sensors using identical printing, drying, and post-treatment conditions as the initial devices. This recycling–reprinting procedure was repeated for up to three cycles. For each cycle, the residence time of the particles in water (from immersion to magnetic collection) was kept below 1 h.

2. Prolonged water-immersion (particle aging/corrosion window) experiment. To assess the effect of extended water exposure during recycling, Fe/Fe$_3$O$_4$ microparticles were intentionally immersed in DI water for predefined durations (1, 3, 5, 10, and 15 days) under ambient conditions. After the specified immersion time, the particles were magnetically collected, rinsed with DI water, and dried to constant mass. Inks were prepared by mixing the aged particles with NaCMC aqueous solution using the same formulation and mixing procedure as for the pristine ink. Sensors were reprinted under identical conditions, and their MR performance was evaluated as described in the MR measurement section.

## Encapsulation of the printed biodegradable magnetoresistive sensors

Several biodegradable encapsulation materials were tested to protect printed biodegradable sensors and regulate their longevity. Ecoflex 00-35-part A and part B were mixed homogeneously in a 1:1 weight ratio and degassed in a vacuum chamber. Gelatin-glycerol gel was prepared by mixing glycerol with water at an 8:2 weight ratio, then dissolving gelatin powder (1:9 ratio) in the mixture, followed by heating at 70 °C under magnetic stirring until a clear viscous solution was obtained. After vacuum drying to remove water, the mixture was kept warm until use. Beeswax was heated to 90 °C for melting. Cellulose acetate was used as acetone solution by 30% weight ratio. Encapsulation materials were drop-cast onto the printed sensors on FFC to form protective layers. To evaluate the environmental stability under different storage conditions, encapsulated and bare sensors were stored under (i) ambient humidity at room temperature, 50 °C, and 80 °C, (ii) 90% relative humidity at room temperature, and (iii) immersion in water at room temperature, 50 °C, and 80 °C. After the designated storage time, the MR performance of the sensors was measured repeatedly at room temperature.

## Biocompatibility characterization

Cell culture: The L929 cell line was purchased from Leibniz Institute DSMZ-German Collection of Microorganisms and Cell Cultures. L929 Cells were cultured in Dulbecco's modified Eagle's medium (DMEM, Gibco) containing 10% fetal bovine serum (Sigma-Aldrich), 2 mM L-glutamine (Sigma-Aldrich) and 1% penicillin/streptomycin (Biochrom). Cells were incubated under 5% $CO_2$ at 37 °C. Cell culture medium was exchanged every 2–3 days.

MTT proliferation assay: Printed sensor pieces ($Fe/Fe_3O_4$-NaCMC) were presterilized under UV light for 4 h, and then the presterilized pieces were added into cell culture media to form a solution of 500 μg· $ml^{-1}$ for L929 cells culturing. The L929 cells were plated in 96-well plates at a density of $5 \times 10^3$ per well, and incubated with or without material samples. After 24, 48, and 72 h of culturing, the cell culture media were replaced with 90 μL of cell culture media and 10 μL of MTT solution (5 mg $ml^{-1}$, Sigma-Aldrich) and then incubated for 2 h in a humidified atmosphere with 5% $CO_2$ at 37 °C. The supernatant was replaced with 50 μl of 100% dimethyl sulfoxide (DMSO, Sigma-Aldrich), followed by shaking for 10 min. An additional 150 μl of 100% DMSO was then added and mixed thoroughly. The data were recorded using a multimodal microplate reader (Cytation 5), measuring optical density at wavelengths of both 570 nm and 690 nm. Three parallel control samples were used for each group. Cell viabilities are equal to the absorbance ratio of the experimental group and the control group.

Cell morphology and immunofluorescent staining: The F-actin and nuclei were stained with phalloidin (conjugated with Alexa Fluor 488, ThermoFisher, 1:200) and Hoechst 33258 (ThermoFisher, 1:10), respectively. Precisely, L929 cells were precultured with and without $Fe/Fe_3O_4$-NaCMC samples (500 μg· $ml^{-1}$) for 24, 48, and 72 h in 8-well μ-slides. After washing with PBS three times, PBS containing 4% paraformaldehyde was used to fix the cells for 10 min. Following permeabilization with 0.2% Triton X-100 for 10 min, the cells were blocked using 3% BSA for 2 h. Phalloidin and Hoechst 33258 were added for 30 min and 10 min incubation at room temperature, respectively. Finally, the antifade reagent Prolong (Thermo Fisher) was added to each well to prevent fluorescence quenching. The cells were imaged using a confocal fluorescence microscope (Olympus IX83).

## Printed sustainable magnets

Glycerol was mixed with water at a 2:1 weight ratio, followed by the addition of gelatin at a 1:9 weight ratio. The mixture was heated to 70 °C under magnetic stirring until a clear viscous solution was obtained[57]. Biocompatible strontium ferrite powders were then added at a 7:3 weight ratio and mixed thoroughly. The resulting ink was cast into molds and cured at room temperature for 24 h to form the final sustainable magnets (Supplementary Fig. 26). The printed magnets were magnetized by a constant magnetic field (2 T) for 5 min using an electromagnet.

## Application of disposable magnetoresistive sensors

(1) Smart gift box: A $Fe/Fe_3O_4$-NaCMC sensor was printed on a paper substrate and integrated into a hybrid electronic circuit (featuring an amplification cascade and a light emitting diode (LED), as detailed in our previous work[21]). A printed biodegradable magnet attached to the cap of the box modulated the resistance of the sensor. The variation in resistance controlled the transistor circuit in the amplifier circuit, which regulated the operation of the LED.

(2) Smart Medicine Box: A printed biodegradable magnet was affixed to the medicine box cap, while a $Fe/Fe_3O_4$-NaCMC sensor with four-point molybdenum electrodes was printed on a NaCMC substrate. Resistance changes were recorded using a Tensormeter (HZDR Innovation GmbH, Germany) with NI LabVIEW software (version 2019 SP1, National Instruments, USA).

(3) On-nail wearable electronics for interactive human-machine interface: Two $Fe/Fe_3O_4$-NaCMC sensors were printed on the fingernails of the index and middle fingers, with contacts integrated beneath the printed layer. The sensors were encapsulated with a gelatin-glycerol gel. A magnetic ring worn on the thumb provided the magnetic field source. MR variation signals were acquired using a Tensormeter and processed with NI LabVIEW software (version 2019 SP1). A custom 3D game programmed in Spline interpreted the sensor signals; when a fingernail approached the magnetic ring, the resulting MR change (exceeding a preset threshold) triggered specific actions (e.g., move forward for the index finger and jump for the middle finger).

## Ethics statement

We applied a permanent magnet to a finger of a user in several studies reported in this manuscript. This magnet is not an electronic component. Furthermore, in several studies, a sensor is applied on skin of a user. These studies are done according to the ethical approval #SR-EK-459122024 from the ethics committee at the Technical University of Dresden. For this case, we have a written consent of the user (one volunteer, male, 29 years old), who was wearing this sensor. The sensor is not worn on the skin for any extended duration.

## Data availability

All of the data supporting the conclusions are available within the article and the Supplementary Information. Additional data are available from the corresponding authors upon request. Source data are provided with this paper.

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

## Acknowledgements

We thank Dr. Tetiana Voitsekhivska (HZDR) for her support with the large-area printing trials, Pavlo Makushko (HZDR) for support with transport characterization, Dr. Denise Erb (HZDR) and Dr. Xiaoxiao Sun (HZDR) for their support with structural characterization, Thomas Schumann (HZDR) for providing the reference Fe thin film, Andreas Worbs (HZDR) for TEM specimen preparation, and Dr. Stanislav Avdoshenko (Leibniz IFW Dresden) for his input on interactive demonstrators at the initial stages of the project. Support by the Structural Characterization Facilities Rossendorf at the Ion Beam Center (IBC) at the HZDR is greatly appreciated. This work is financed in part via the European Commission HORIZON RIA (project REGO; ID: 101070066; D.M.), ERC grant 3DmultiFerro (Project number: 101141331; D.M.), and German Research Foundation (DFG grant # MA5144/37-1; D.M.). Furthermore, L.B. acknowledges the financial support via the ERC grant Immunochip (Project number: 101045415) and German Research Foundation (Deutsche Forschungsgemeinschaft, DFG) via the "Responsible Electronics in the Climate Change Era – REC²" Cluster of Excellence (EXC 3035, Project-ID 533607596). O.G. acknowledges the financial support by the Deutsche Forschungsgemeinschaft (DFG, German Research Foundation), Project ID no. 405553726, TRR 270. R.L. acknowledges the financial support by National Natural Science Foundation of China, Project ID: U24A6001 and 52127803. L.G., X.W., and Y.L. acknowledge the China Scholarship Council (CSC) for the PhD scholarship. P.T.D acknowledges the financial support via the project FlexiMMG (project number 101106524) of Horizon-MSCA-2022-PF-01 (European Commission). Furthermore, the funding of TEM Talos by the German Federal Ministry of Education and Research (BMBF, grant No. 03SF0451) in the framework of HEMCP is acknowledged.

## Author contributions

D.M., R.X., and L.G. conceived the idea of eco-sustainable magnetoelectronics. D.M. supervised the project. L.G. fabricated sensors and performed their structural, electrical, and magnetoresistive characterization with the support of D.M., R.X., E.S.O.M., P.T.D., X.W., Y.Z., Q.Z., and R.L. Printing trials were carried out by L.G., Y.Z., S.G., and X.W. The demonstrators were designed and realized by L.G., E.S.O.M., X.B., and S.L. with support from X.W., R.X., Y.Z., R.L., and D.M. R.H. carried out the TEM-based analyses. F.G. and I.V. performed XRD measurement and data analysis. Y.L. carried out Raman measurements and data analysis. X.P., Z.J., and L.B. carried out the biocompatibility studies. C.V. and S.M performed rheology characterization. L.G. developed the biocompatible magnets with the support of D.M. and O.G. Micromagnetic simulations were performed by O.V.P. The mechanism of magnetoresistance and magnetic flux guide was developed by L.G., D.M., P.T.D., O.V.P., and R.L. The manuscript was written by L.G., R.X., P.T.D., and D.M., with comments from all authors. All co-authors edited the manuscript.

## Funding

## Competing interests

The authors declare no competing interests.
