## [Transparent Peer Review file · Nature Communications]

Eco-sustainable magnetoresistive sensors towards disposable magnetoelectronics

Corresponding Author: Dr Denys Makarov

Version 0:

Reviewer comments:

Reviewer #1

(Remarks to the Author)

This is a timely and technically strong manuscript from Makarov and co-authors. The demonstration of biodegradable, recyclable magnetoresistive sensors with credible low-field performance is an important contribution to sustainable magnetoelectronics. The work is clearly of interest and should be published at Nature Communications; however, some major points require further explanation and contextualisation. Please see individual comments below:

- Particle recovery is shown, but it is not very clear whether the reclaimed Fe/Fe₃O₄ retains functional quality. Even a discussion of expected reusability and potential barriers would help support the circularity claim.
- The issue of water sensitivity is acknowledged, but the implications for storage may need further clarification. The authors should discuss the operational window e.g. humidity, temperature, and time in service and clarify how encapsulation strategies change that window.
- The authors should explain how particle size distribution and orientation affect the effective gain factor in the printed composite.
- How does sensitivity and figure of merit at <10 mT compare to samples from the literature?
- Was sensor readout stable across practical bias ranges and was there an optimal operating voltage to minimize drift?
- Finally, since many binders were used, is there any binder feature (e.g. polarity or mechanical performance) that heavily influences the magnetic or electrical coupling between Fe/Fe₃O₄ and the overall MR response?

Reviewer #2

(Remarks to the Author)

The study demonstrates clear potential for significant contributions to the field. However, several substantive issues require careful revision and clarification before this work can be considered for publication:

1. This manuscript systematically optimizes the magnetic and electronic properties (oxidation temperature, particle size, and binder type) which is appreciative. However, the authors made a major point in using "industry-scale screen printing" (Fig. 1b, Supplementary Fig. 1) for high-throughput fabrication. This is a key claim in terms of scalability and practicality. Nonetheless, they provide no data regarding the optimization of ink to prove its printability. To that end, it is important to know the rheological profile of the ink. Rheological data (viscoelastic properties, shear-thinning behavior of the ink for screen printing, G'/G'' behavior etc., and contact angle measurements) are critical parameters. In addition, answering the reason for selecting the 60% volume ratio would show how these choices affect the printability of the ink. The manuscript would benefit from the inclusion of such characterization of the ink and aid in the reproducibility of the work.

2. On page 6, authors mentioned "These designs work in concert, delivering exceptional low-field response, as evidenced

by a higher magnetoresistance ratio at 10 mT relative to all printed magnetoresistive sensors reported previously^{21–31,40} (Fig. 1h)". Authors are requested mention the supporting supplementary table.

3. In Figure 1h, the authors highlight the large-scale fabrication, green synthesis, biocompatibility, biodegradability, recyclability, and %MR at 10 mT to emphasize the practical significance of their sensors. However, it is unclear which specific synthesis parameter(s) was/were considered in this comparison. Based on the figure caption, it seems only the binders were compared. The authors are requested to clarify the parameters used and justify their selection to ensure that the comparison is scientifically meaningful and is not limited to a single fabrication variable.

4. The authors use an acid wash (Vitamin C) treatment for the selective removal of the Fe₃O₄ shell and demonstrate a transition to linear current-voltage behavior, indicative of ohmic conduction, whereas the printed Fe/Fe₃O₄ samples exhibit nonlinear current-voltage characteristics, which they attribute to the variable-range hopping mechanism (Supplementary Fig. 5). In addition, on page 9, the authors claim that "The magnetoresistive behavior of Fe/Fe₃O₄ core-shell particle sensors can be attributed to spin-dependent hopping across magnetically disordered interfaces between adjacent Fe₃O₄ shells. Upon application of an external magnetic field, the alignment of local magnetic moments reduces spin disorder and enhances carrier hopping mobility, giving rise to negative magnetoresistance effects [Ref. 47,48]." Authors are requested to include the relevant MR curve (if possible) for a direct and robust visual demonstration of the indispensable role of the magnetically disordered interfaces between adjacent Fe₃O₄ shell in generating the magnetoresistance.

5. The authors claimed an improved low-field sensitivity with exceptionally reliable practical functionality. However, a comparison of devices produced within the same batch and different batches would strengthen the scientific rigor of the technology. The authors are requested to provide a statistical analysis of the key performance parameters based on measurements from multiple independent sensor devices to ensure reproducibility. The Methods section should explicitly state the sample size (n) for each dataset.

6. On page 11, the authors have successfully demonstrated the optimization of device performance by varying the core diameter and shell thickness using numerical simulations. The authors also mentioned "..... as the core diameter increases, the greater permeability contrast improves flux concentration. However, beyond a critical size, the resulting flux distribution reaches a geometric limit, no longer enhancing the field in grain boundaries." The observed normalized MR shown in Figure 4i is related to the overall particle size. The authors are required to provide direct microstructural evidence (e.g., cross-sectional SEM/TEM/HRTEM) to quantify the final core diameter and precise shell thickness of the particles. These values should be explicitly stated in the main text and correlated with the performance data.

7. The claim of a "truly closed-loop sustainable life cycle" (e.g., Page 12, Fig. 5a) should be reconsidered. This phrase implies a rigorous Life Cycle Assessment (LCA) that quantitatively evaluates environmental impacts from raw material extraction to end-of-life. The authors are encouraged to reflect on the presented evidence more accurately. For instance, wording such as "demonstrates key attributes for a sustainable life cycle" or "presents a viable pathway towards closed-loop sustainability" would be more appropriate. Moreover, including a qualitative discussion acknowledging that a comprehensive quantitative LCA would be a valuable future direction could add important nuances and strengthen the environmental relevance of this study.

Minor comments:

i) The placement of Figure 1c in the current version may be confusing for readers. The authors are encouraged to rearrange the figure to ensure a more logical flow and clearer distinction from Figures 1d and 1e. Adding labels such as "Fe core (conductor & flux guide)" and "Fe₃O₄ shell (MR-active)" to the schematic would make the synergistic mechanism immediately clearer.

ii) In Supplementary figure 11, the colors described in the caption do not correspond to those of the respective curves in the graph, particularly the green curve in S11-a. The authors are requested to verify this inconsistency.

iii) Page 8, section "Spin-dependent hopping-mediated grain boundary" — The sentence "..., the Fe/Fe₃O₄ sensors exhibit resistance reduction by over two orders of magnitude..." should cite the specific supplementary table where these data are presented (apparently referring to Supplementary Tables 3). The authors are advised to verify this reference.

Reviewer #3

(Remarks to the Author)

Version 1:

Reviewer comments:

Reviewer #1

(Remarks to the Author)

The authors have successfully addressed all of my queries, and I find the current version suitable for publication in Nature Communications.

Reviewer #2

(Remarks to the Author)

I thank the authors for their careful revision of the manuscript and for the detailed responses to the reviewers' comments. The concerns raised during the previous round of review have been satisfactorily addressed, and I have no further suggestions.

Reviewer #3

(Remarks to the Author)

Response letter

We thank the Reviewers for their constructive comments, which we used to refine our manuscript. All changes are indicated in the manuscript and supporting information files.

Our itemized responses to the remarks of the Reviewers are summarized below.

Reviewer #1 (Remarks to the Author):

Overall Comment: *This is a timely and technically strong manuscript from Makarov and co-authors. The demonstration of biodegradable, recyclable magnetoresistive sensors with credible low-field performance is an important contribution to sustainable magnetoelectronics. The work is clearly of interest and should be published at Nature Communications; however, some major points require further explanation and contextualisation. Please see individual comments below:*

Response: We sincerely thank the Reviewer for this positive and encouraging assessment of our manuscript. We have revised the manuscript to strengthen clarity, expand the relevant discussion, and better position our results within the broader literature. We address each individual comment in detail below, and all corresponding changes have been incorporated into the revised manuscript.

Comment 1: - *Particle recovery is shown, but it is not very clear whether the reclaimed Fe/Fe₃O₄ retains functional quality. Even a discussion of expected reusability and potential barriers would help support the circularity claim.*

Response: To explicitly address whether the reclaimed Fe/Fe₃O₄ microparticles retain functional quality, we have added a dedicated recyclability assessment of the recovered particles by re-formulating the ink and reprinting sensors using Fe/Fe₃O₄ fillers collected after repeated recycling cycles. Under practical recycling conditions, where the particle residence time in water is kept below 1 h per cycle (NaCMC dissolution followed by magnetic collection), the MR curves of the reprinted sensors nearly overlap with those of the initial devices, and the MR ratio at 100 mT remains essentially unchanged over three consecutive recycling cycles (new Supplementary Fig. 4a,b). These results demonstrate that the reclaimed Fe/Fe₃O₄ particles maintain their functional performance and support the reusability required for circular operation.

In addition, we now discuss potential barriers and practical constraints. Because water is used both as the recycling medium and as the ink solvent, prolonged water exposure can gradually corrode/age the particles and lower the MR response. We therefore performed a worst-case “extended immersion” experiment, intentionally prolonging the Fe/Fe₃O₄ particle immersion time during a single recycling step, which leads to a gradual decrease of the MR response, the MR ratio (at 100 mT) reaching 2.57% after 15 days (new Supplementary Fig. 4 c,d). This identifies an operational boundary and provides guidance for fabrication and recycling: the ink should be prepared on demand, and the recycling process should be conducted promptly to minimize the particle residence time in water. Importantly, since the performance degradation becomes evident only on a multi-day timescale, while practical recycling is completed within <1 h per cycle, we expect that a reasonable number of recycling loops will not cause an obvious performance loss in reprinted sensors.

Revisions:

Supporting Information:

Added new Supplementary Fig. 4:

Supplementary Fig. 4. Effect of the recycling procedure on the MR performance of reprinted sensors. **a.** MR curves of sensors printed with pristine Fe/Fe₃O₄ microparticles (“Initial”) and with particles recovered after 1st–3rd recycling cycles. In each recycling cycle, the NaCMC binder was dissolved in water and the dispersed Fe/Fe₃O₄ particles were collected magnetically; the particle residence time in water was kept below 1 h. **b.** Corresponding MR ratio at 100 mT as a function of recycling cycle. **c.** MR curves of sensors reprinted using Fe/Fe₃O₄ particles exposed to water for different immersion durations (particle corrosion time) during a single recycling step (1–15 days), showing progressive performance degradation upon prolonged immersion. **d.** Corresponding change of the MR performance as a function of the immersion (corrosion) time.

The recycling of the printed sensors relies on the water solubility of the NaCMC binder. During end-of-life processing, the printed composite disintegrates in water as NaCMC dissolves, releasing Fe/Fe₃O₄ particles into the suspension. Owing to their intrinsic magnetism, the particles can be efficiently collected using a permanent magnet, dried, and re-dispersed in a freshly prepared NaCMC aqueous solution to formulate the ink for reprinting. This water-enabled dissolution–magnetic collection route constitutes a key step enabling closed-loop recyclability without harsh reagents.

Importantly, when the recycling step is performed promptly, the MR performance of reprinted sensors remains essentially unchanged. As shown in panels **a** and **b**, particles recovered through three consecutive recycling cycles, each with a short water exposure time (< 1 h), yield reprinted sensors whose MR curves nearly overlap with that of the initial device, and the MR ratio at 100 mT stays around 3%. This indicates a good reusability of the Fe/Fe₃O₄ particles under practical, time-efficient recycling conditions.

In contrast, a prolonged water immersion leads to a gradual decline in the MR performance (panels **c** and **d**). When Fe/Fe₃O₄ particles are kept in water for extended durations during a single recycling step,

the MR performance gradually decreases, reaching 2.57% after 15 days of immersion, due to the water induced corrosion.

These observations provide practical guidance for both fabrication and recycling. First, because the ink uses water as the solvent and the particles are continuously exposed to an aqueous environment once mixed, prolonged storage can induce ink aging and performance loss. Therefore, the ink should be prepared on demand (i.e., mix NaCMC aqueous solution with Fe/Fe₃O₄ particles shortly before printing). Second, during recycling, the residence time of particles in water should be minimized. Notably, since the performance degradation becomes evident only after multi-day immersion while typical recycling requires less than 1 h per cycle, water-induced corrosion is a relatively slow process in comparison to the practical recycling timescale. Consequently, a reasonable number of recycling loops is expected to have no impact on the MR performance of reprinted sensors, supporting the feasibility of closed-loop reuse of the MR fillers.

Main text: page 7

added:

Notably, under practical recycling conditions (water exposure < 1 h per cycle), the reclaimed Fe/Fe₃O₄ microparticles retain their functional quality, as evidenced by the essentially unchanged MR performance of reprinted sensors over multiple recycling cycles (Supplementary Fig. 4), confirming the excellent reusability of the magnetic fillers.

Method: Page 19

added:

Recyclability and water-immersion aging tests of reclaimed Fe/Fe₃O₄ microparticles

1. Recycling–reprinting cycle (practical recycling condition). Printed sensors were recycled by exploiting water solubility of the NaCMC binder. Discarded sensors were immersed in deionized (DI) water until the printed layer disintegrated and the composite dispersed. The released Fe/Fe₃O₄ microparticles were collected from the suspension using a permanent magnet, rinsed with DI water to remove residual NaCMC, and dried under ambient conditions (or under mild vacuum) until constant mass. The reclaimed particles were then re-mixed in a freshly prepared NaCMC aqueous solution to formulate inks following the same protocol as for the pristine ink. The ink was subsequently reprinted to fabricate new sensors using identical printing, drying, and post-treatment conditions as the initial devices. This recycling–reprinting procedure was repeated for up to three cycles. For each cycle, the residence time of the particles in water (from immersion to magnetic collection) was kept below 1 h.

2. Prolonged water-immersion (particle aging/corrosion window) experiment. To assess the effect of extended water exposure during recycling, Fe/Fe₃O₄ microparticles were intentionally immersed in DI water for predefined durations (1, 3, 5, 10, and 15 days) under ambient conditions. After the specified immersion time, the particles were magnetically collected, rinsed with DI water, and dried to constant mass. Inks were prepared by mixing the aged particles with NaCMC aqueous solution using the same formulation and mixing procedure as for the pristine ink. Sensors were reprinted under identical conditions, and their MR performance was evaluated as described in the MR measurement section.

Comment 2: - *The issue of water sensitivity is acknowledged, but the implications for storage may need further clarification. The authors should discuss the operational window e.g. humidity, temperature, and time in service and clarify how encapsulation strategies change that window.*

Response: In the original manuscript, we primarily evaluated a long-term water-immersion stability at room temperature (up to 90 days) to demonstrate that biodegradable encapsulation can regulate the water-triggered performance degradation (Fig. 2c). To clarify practical storage/service implications, we have now performed a systematic aging study under controlled humidity and temperature conditions and added the results as Supplementary Fig. 5.

The new data define the operational window as follows:

1. Ambient humidity (room temperature to 80 °C): Bare sensors and those encapsulated with beeswax or ecoflex show essentially unchanged MR over 20 days, indicating that for dry/ambient environments, encapsulation is not required for at least 20 days even at elevated temperature (≤ 80 °C). Notably, beeswax is not suitable for high-temperature encapsulation because it melts at ~ 80 °C.
2. High humidity (95% RH, room temperature): The unencapsulated devices become non-conductive rapidly (within ~ 2 h; the first scheduled measurement already fails), whereas both beeswax and ecoflex encapsulation preserve stable MR for at least 20 days, demonstrating that encapsulation effectively shifts the operational window to highly humid environments.
3. Water immersion (50 °C and 80 °C): Under hot water exposure, the lifetime becomes strongly encapsulation-dependent. Beeswax provides only short-term protection at 50 °C (stable in the first ~ 5 days, followed by pronounced degradation and device failure by day ~ 15). In contrast, ecoflex offers the broadest window, maintaining stable MR for at least 20 days at 50 °C in water, and for ~ 15 days at 80 °C before failure at day 20.

Overall, these results complement the original long-term room-temperature immersion data (Fig. 2c) and explicitly show how encapsulation expands the usable humidity/temperature regime—from dry ambient storage to high-humidity and warm-water environments—allowing application-specific tailoring of device lifetime via the choice of encapsulation layer.

Revision:

Added new Supplementary Fig. 5:

Supplementary Fig. 5. Effect of encapsulation and storage conditions on MR stability of printed Fe/Fe₃O₄-NaCMC sensors. MR ratio (at 100 mT) of printed sensors without encapsulation (“Bare”) and with beeswax or Ecoflex encapsulation after storage under **a.** ambient humidity at room temperature, **b.** ambient humidity at 50°C, **c.** ambient humidity at 80°C, **d.** 95% relative humidity at room temperature, **e.** immersion in water at 50°C, and **f.** immersion in water at 80°C. Beeswax was not tested at 80°C due to melting. MR was measured at room temperature after the indicated storage time. Data points at 0% correspond to the loss of electrical conduction (device failure).

The printed Fe/Fe₃O₄-NaCMC composite is intrinsically moisture-sensitive because the NaCMC matrix is hydrophilic and water-soluble; moisture uptake can swell/soften the binder and disrupt the particle percolation network, leading to a rapid increase in resistance and eventual loss of conductivity. The sensors remain stable for at least 20 days under ambient humidity even when continuously stored at up to 80°C, whereas unencapsulated devices fail rapidly at 95% relative humidity (RH). The encapsulation effectively delays the moisture ingress and thus expands the operational window: beeswax and Ecoflex both protect the device in high humidity at room temperature, while under water immersion the lifetime becomes temperature and encapsulation dependent. Beeswax provides only short-term protection at elevated temperatures (accelerated degradation in warm water and melting at ~80°C), whereas Ecoflex offers the widest window (stable for ≥20 days in water at 50°C and ~15 days at 80°C). Together with the long-term room-temperature immersion results in the Fig. 2c (main text), these data highlight that the service lifetime can be application-specifically tuned by selecting an appropriate biodegradable encapsulation strategy.

Main text: page 8

added a brief discussion:

Consistent with these immersion results, accelerated aging under controlled humidity and temperature further confirms that encapsulation markedly expands the operational window from ambient conditions to high-humidity and warm-water environments (Supplementary Fig. 5).

Method: Page 20. Section of Encapsulation of the printed biodegradable magneto-resistive sensors

Added:

To evaluate the environmental stability under different storage conditions, encapsulated and bare sensors were stored under (i) ambient humidity at room temperature, 50°C, and 80°C, (ii) 90% relative humidity at room temperature, and (iii) immersion in water at room temperature, 50°C, and 80°C. After the designated storage time, the MR performance of the sensors was measured repeatedly at room temperature. ~~The encapsulated sensors were immersed in water at room temperature to assess stability, and their MR performance was measured periodically.~~

Comment 3: - *The authors should explain how particle size distribution and orientation affect the effective gain factor in the printed composite.*

Response: The gain factor is defined as the amplification of the local magnetic field at the shell–shell grain-boundary junctions by the Fe core (i.e., the ratio between the locally concentrated field and the applied external field). In a printed composite, this amplification is determined by many neighboring-particle junctions along percolation paths. Therefore, both particle size distribution (size mismatch between adjacent particles) and particle orientation (relative angle between the external field and the percolation path) can affect the effective gain factor.

Following the reviewer’s suggestion, we performed additional simulations (external field 100 mT) to quantify these effects while fixing the Fe₃O₄ shell thickness to 10 nm. To evaluate the influence of size distribution, we considered two contacting core–shell particles, fixed the core radius of one particle at 120 nm, and varied the core radius of the second particle from 50 to 120 nm (new Supplementary Fig. 16). The gain factor at the junction points changes only moderately, remaining in the range of ~13–16, indicating that reasonable size mismatch introduces only a limited variation in local field amplification. This behavior is related to the change in magnetization distribution near the junction, where the smaller particle tends to be more uniformly magnetized under the applied field.

We also examined the influence of field orientation by rotating the external magnetic field with respect to the symmetry axis of the contacting particle pair (Supplementary Fig. 17 and 18). For two identical particles, the gain factor is highest when the field is aligned with the interparticle axis and decreases progressively with increasing tilt, reaching values of ~3 for the perpendicular orientation. A size-mismatched pair (core radii 120 nm and 60 nm) follows the same overall trend and shows a more monotonic angular dependence (new Supplementary Fig. 18), consistent with a more homogeneous magnetization of the smaller particle. It is worth noting that the gain factor is highly sensitive to the relative orientation between the external magnetic field and the percolation path connecting adjacent particles (Supplementary Fig. 17). As the alignment shifts from parallel to tilted, the gain factor decreases progressively. Due to the random arrangement of particles within the composite, the overall averaged effect is expected to be lower, which is in accordance with the experimentally observed gain factor of approximately 10.

Overall, these simulations indicate that particle size distribution introduces only a modest spread in the local gain factor, whereas orientation can cause a stronger reduction at individual junctions. In a randomly packed printed composite, averaging over many junctions with different orientations is therefore expected to yield an effective gain factor lower than the ideal aligned-pair maximum, consistent with the experimentally observed value around 10.

Revision:

Added new **Supplementary Fig. 16 and 18.**

Supplementary Fig. 16. Simulation of particle-size-mismatch effects on the local gain factor at a core-shell junction. a-e. Spatial maps of the gain factor (colormap) in the axial cross-section of two

contacting spherical Fe/Fe₃O₄ core-shell particles with different core-radius ratios. The larger particle is kept with a constant core radius, R_1 , of 120 nm, while the smaller particle has core radii, R_2 , of 50 nm (a), 60 nm (b), 80 nm (c), 100 nm (d), and 120 nm (e). In all cases, the Fe₃O₄ shell thickness is fixed at 10 nm. (f) Gain factor evaluated at the junction as a function of the core-radius ratio (R_2/R_1), showing only a modest variation across the investigated size-mismatch range.

Supplementary Fig. 18. Simulation of field-orientation dependence of the local gain factor in a size-mismatched Fe/Fe₃O₄ particle pair. a-c. Spatial maps of the local gain factor (colormap) in the x - z axial cross-section of two contacting spherical core-shell Fe/Fe₃O₄ particles (core radii: 120 nm and

60 nm; shell thickness: 10 nm), together with the corresponding equilibrium magnetization distribution (arrows). An external magnetic field of 100 mT is applied with a tilt angle $\theta = 30^\circ$ (panel **a**), 60° (panel **b**), and 90° (panel **c**) relative to the horizontal/interparticle axis. **d.** Gain factor at the junction as a function of θ , showing a progressive reduction with increasing tilt, consistent with the evolution of the equilibrium magnetic state near the junction.

Main text: page 11

Revised discussion:

Additional simulations further indicate that a reasonable size mismatch between two contacting particles introduces only a modest variation in the local gain factor (Supplementary Fig. 16), whereas the gain factor is highly sensitive to the relative orientation between the external magnetic field and the percolation path connecting adjacent particles (Supplementary Fig. 17 and 18).

Comment 4: - *How does sensitivity and figure of merit at <10 mT compare to samples from the literature?*

Response: A detailed quantitative comparison is provided in Supplementary Table 1, where sensitivity and FoM values at ≤ 10 mT are compiled from representative printed magnetoresistive sensors. To further improve clarity and visibility, we additionally prepared a radar chart summarizing the sensitivity and FoM in the low-field regime (< 10 mT) (Fig. R1). Based on this comparison, the magnetoresistance ratio at 10 mT of our Fe/Fe₃O₄-based sensor is higher than that of all previously reported printed magnetoresistive sensors. In terms of low-field sensitivity (< 10 mT), our sensor outperforms the majority of reported printed systems, typically by several-fold and, in some cases, approaching one order of magnitude. An exception is a printed Ni₈₁Fe₁₉ permalloy-based sensor (DOI: 10.1038/s41467-022-34235-3), which exhibits very high sensitivity due to the ultrahigh magnetic permeability and low coercivity of permalloy. Notably, this Ni₈₁Fe₁₉-based sensor shows a relatively low magnetoresistance ratio ($< 1\%$) and saturates at magnetic fields below ~ 2.5 mT, thereby limiting both signal amplitude and effective detection range. In addition, Ni₈₁Fe₁₉ relies on a high nickel content ($> 80\%$), which involves hazardous and non-eco-sustainable elements. A similar trend is observed for the figure of merit at < 10 mT. Our sensor exhibits a FoM that is higher than most reported printed magnetoresistive sensors, often by more than an order of magnitude, with only two Ni₈₁Fe₁₉-based systems (DOI: 10.1038/s41467-022-34235-3 and 10.1002/adma.202005521) showing higher values. Overall, this comparison highlights that the proposed Fe/Fe₃O₄-based sensors achieve a rare combination of high low-field performance, wide detection range, and intrinsic eco-sustainability.

Fig. R1. Low field (<10 mT) performance (MR, sensitivity and FoM) comparison with the literature-reported printed MR sensors. The Ref. numbers correspond to the ones in the main text.

Comment 5: - Was sensor readout stable across practical bias ranges and was there an optimal operating voltage to minimize drift?

Response: We evaluated sensor readout stability and identified a practical operating bias by combining (i) time-domain output stability, (ii) signal-to-noise ratio (SNR) and noise spectral density (NSD) under different excitation currents, and (iii) bias-dependent MR characteristics (Fig. R2a–d).

From the device geometry, the optimized Fe/Fe₃O₄ sensor yields an electrical resistivity of $\sim 0.6 \Omega \cdot m$, corresponding to a channel resistance of $\sim 560 \Omega$. Accordingly, the excitation-current range in Fig. R2b (283 nA–286 μA) maps to a bias-voltage range of $\sim 0.16 mV$ – $0.16 V$, with ultralow power dissipation of $\sim 45 pW$ – $46 \mu W$, which avoids self-heating. At a representative practical operating point ($\approx 0.16 V$), the time-domain output remains stable over the measurement window with no observable drift (Fig. R2a). The NSDs measured at room temperature show that very low excitation currents lead to an elevated noise floor, while increasing the excitation current suppresses the NSD and finally yields a saturated and stable baseline around 0.1 V bias voltage ($\sim 133 \mu A$ excitation current) (Fig. R2b). Consistently, at an ultralow bias of 1 mV ($\approx 1.8 \mu A$ for a $\sim 560 \Omega$ channel), the MR curve becomes noise-dominated and the apparent MR is limited by the poor SNR (Fig. R2c).

We further quantified the MR response as a function of bias voltage over 0.1–5 V (Fig. R. 2d). From Fig. R2d, the MR performance is found to be stable across this bias range. It is important to note that higher biases (>1 V) cause a measurable MR suppression (we observed a decrease from ~3.3% to 2.8% MR between 0.1 V and 5 V), consistent with the voltage-induced reduction of the hopping barrier. Therefore, we recommend (and used in this work) a low-bias operating point around 0.1 V combined with low-noise AC (lock-in) readout, which minimizes noise spectral density while maintaining sensitivity near its maximum.

Fig. R2. **a.** Time-domain output stability. **b.** Noise spectral density (NSD) of the sensor at different excitation currents. **c.** Low signal-to-noise ratio MR curve measured at 1mV bias voltage. **d.** Voltage-dependent magnetoresistance characterization.

Comment 6: - Finally, since many binders were used, is there any binder feature (e.g. polarity or mechanical performance) that heavily influences the magnetic or electrical coupling between Fe/Fe₃O₄ and the overall MR response?

Response: In our experiments, we intentionally tested a diverse set of polymeric binders, including alginate, starch, egg white, and NaCMC, which span a wide range of chemical polarity, functional groups, and mechanical characteristics (Fig. 2d and Supplementary Fig. 6). Despite these pronounced differences, the resulting sensors exhibit very similar magnetoresistance ratios (around -3%) and sensitivities ($> 3.5 \text{ T}^{-1}$), without any systematic trend linked to binder chemistry or mechanics. This observation indicates that no specific binder feature, such as polarity or mechanical compliance, heavily dominates the magnetic or electrical coupling in the composite. All tested binders are electronically insulating and magnetically inert. Therefore, they do not introduce additional electronic or magnetic coupling pathways. Instead, the binder primarily serves to immobilize the Fe/Fe₃O₄ core-shell microparticles and stabilize the percolated microstructure. Consistent with the transport analysis presented in Fig. 3, the overall MR response is governed by spin-dependent hopping across Fe₃O₄ grain boundaries and by the Fe-core-induced magnetic flux guiding effect, both of which are intrinsic to the Fe/Fe₃O₄ architecture rather than the binder. The weak dependence on binder chemistry further highlights the robustness and process tolerance of the proposed sustainable sensor platform.

Reviewer #2 (Remarks to the Author):

Overall Comment: *The study demonstrates clear potential for significant contributions to the field. However, several substantive issues require careful revision and clarification before this work can be considered for publication:*

Response: We sincerely thank the Reviewer for their positive evaluation of our manuscript.

Comment 1: *1. This manuscript systematically optimizes the magnetic and electronic properties (oxidation temperature, particle size, and binder type) which is appreciative. However, the authors made a major point in using “industry-scale screen printing” (Fig. 1b, Supplementary Fig. 1) for high-throughput fabrication. This is a key claim in terms of scalability and practicality. Nonetheless, they provide no data regarding the optimization of ink to prove its printability. To that end, it is important to know the rheological profile of the ink. Rheological data (viscoelastic properties, shear-thinning behavior of the ink for screen printing, G'/G'' behavior etc., and contact angle measurements) are critical parameters. In addition, answering the reason for selecting the 60% volume ratio would show how these choices affect the printability of the ink. The manuscript would benefit from the inclusion of such characterization of the ink and aid in the reproducibility of the work.*

Response: In the revised manuscript, we added a dedicated rheological assessment and clarified the rationale for selecting the 60 vol% Fe/Fe₃O₄ particles formulation (Fe/Fe₃O₄ particles volume fraction in the final dried composite).

Rheology added (viscosity window, shear-thinning, and G'/G''). We performed stepwise steady-shear measurements (1–100 s⁻¹) and small-amplitude oscillatory frequency sweeps for inks with varied NaCMC concentration (10 wt% and 15 wt% NaCMC in the aqueous binder, at 60 vol% Fe/Fe₃O₄ particles) and varied Fe/Fe₃O₄ particles loading (30/60/90 vol% Fe/Fe₃O₄ particles, at 10 wt% NaCMC). The results and detailed discussion are now provided as new Supplementary Fig. 2, demonstrating suitable viscosity window, pronounced shear-thinning and a tunable viscoelastic response (G'/G'') as a function of formulation, consistent with screen-printing-compatible ink behavior.

Role of NaCMC concentration (tuning viscosity/viscoelasticity and ensuring ink stability). NaCMC concentration is a key formulation parameter that simultaneously tunes the viscosity and viscoelasticity of the particle-laden ink, thereby governing both mesh transfer under shear and shape retention after printing. In our screening, decreasing NaCMC concentration systematically reduced the low-shear viscosity and viscoelastic moduli (G' and G''), weakening the polymer/particle network that stabilizes the suspension. When the NaCMC concentration was too low, the ink became unstable and exhibited rapid sedimentation and phase separation at the same particle loading.

Rationale for selecting 60 vol% Fe/Fe₃O₄ particles (printability, electrical percolation, and material efficiency). The choice of 60 vol% Fe/Fe₃O₄ particles was guided by a balanced optimization across processing and device requirements. At low particle loading (e.g., 30 vol%), it is difficult to establish a robust conductive percolation network after printing, whereas excessively high loading produces a highly solid-like paste that can hinder mesh transfer and increase clogging risk. In addition, very high particle fractions reduce the effective binder volume available for particle rearrangement and consolidation during drying, which can slightly compromise magnetoresistance performance (Supplementary Fig. 24) while also increasing material consumption without commensurate benefits. Accordingly, the formulation of 60 vol% Fe/Fe₃O₄ particles provides a practical compromise between reliable screen printing and robust electrical/magnetoresistive performance.

Revision:

Added new Supplementary Fig. 2:

Supplementary Fig. 2. Rheological characterization of NaCMC–Fe/Fe₃O₄ inks for screen printing.

a. Stepwise steady-shear viscosity of inks containing 60 vol% Fe/Fe₃O₄ particles (in the final dried composite) formulated using NaCMC aqueous binders with different NaCMC concentrations (10 wt% and 15 wt% NaCMC in water). The shear rate was sequentially set to 1, 10, 30, and 100 s⁻¹ (values indicated in the panel). **b.** Frequency-dependent storage modulus (G') and loss modulus (G'') of the inks in panel **a** measured by small-amplitude oscillatory shear. **c.** Stepwise steady-shear viscosity of inks formulated with a 10 wt% NaCMC aqueous binder and different Fe/Fe₃O₄ particles volume fractions (30, 60, and 90 vol% in the final dried composite). **d.** Corresponding oscillatory frequency sweeps of G' and G'' for the inks in **c**. NaCMC concentration refers to the polymer weight fraction in the aqueous binder solution used for ink preparation; particles content refers to the Fe/Fe₃O₄ particle volume fraction in the composite after solvent evaporation. All measurements were performed at 22°C using a plate–plate geometry (1 mm gap).

Rheological characterization was conducted to evaluate the printability window of NaCMC–Fe/Fe₃O₄ composite inks by varying the NaCMC concentration in the aqueous binder and the Fe/Fe₃O₄ particles volume fraction in the final dried composite. First, viscosity of the ink is in the range that is suitable for screen printing. The stable formulations exhibit pronounced shear-thinning behavior in stepwise steady-shear tests: the viscosity decreases systematically when the shear rate is increased from 1 to 100 s⁻¹, which is desirable for screen printing because the ink can flow readily through the mesh under the squeegee yet recover a higher viscosity at low shear to suppress spreading after printing. Increasing the NaCMC concentration (10 wt% → 15 wt%, at 60 vol% Fe/Fe₃O₄ particles) substantially raises the low-shear viscosity and the viscoelastic moduli (G' and G''), indicating a strengthened polymer/particle network that improves the suspension stability and shape retention. In contrast, inks formulated with a 5 wt% NaCMC binder at 60 vol% Fe/Fe₃O₄ particles were not stable (rapid sedimentation and phase separation), evidencing that an insufficient binder concentration fails to provide an adequate yield/structural strength for printing.

At the fixed NaCMC concentration (10 wt%), increasing the Fe/Fe₃O₄ particles loading (30 → 60 → 90 vol%) progressively elevates the viscosity and moduli. The 90 vol% Fe/Fe₃O₄ particles ink shows a markedly solid-like response with high, weakly frequency-dependent G' and $G' \gg G''$ across the measured range, consistent with a jammed/percolated particle skeleton. While such a structure can aid shape fidelity, it may also impede mesh transfer and increase the risk of screen clogging. By comparison, 10 wt% NaCMC–60 vol% Fe/Fe₃O₄ particles provides a balanced rheological profile—stable dispersion, strong shear-thinning, and stable viscoelasticity—supporting reliable screen printing with good pattern fidelity and minimal slumping.

Main text: page 5

Rheological measurements further corroborate the screen-printing suitability of the selected ink, featuring the suitable viscosity for screen printing and pronounced shear-thinning under increasing shear rate (Supplementary Fig. 2).

Method: page 15

Rheology characterization

Rheological measurements were performed to evaluate the viscoelastic behaviour of the inks. The MCR 302 rheometer (Anton Paar) equipped with the MRD180/1T magnetocell in a plate–plate configuration (1 mm gap) was used to determine the shear-thinning and oscillatory behaviour of the inks. All measurements were performed at 22°C and the sample volume was constant.

Comment 2: 2. On page 6, authors mentioned “These designs work in concert, delivering exceptional low-field response, as evidenced by a higher magnetoresistance ratio at 10 mT relative to all printed magnetoresistive sensors reported previously^{21–31,40} (Fig. 1h)”. Authors are requested mention the supporting supplementary table.

Response: We agree that the comparison statement on page 6 should be explicitly supported by the corresponding supplementary data. In the revised manuscript, we have updated the sentence to directly reference Supplementary Table 1, which provides a detailed quantitative comparison of printed magnetoresistive sensors reported in the literature. This table underpins the comparison shown in Fig. 1h and substantiates the statement regarding the magnetoresistance ratio at 10 mT. The revised sentence now explicitly cites Supplementary Table 1 to improve clarity and transparency.

Revision (for comment 2):

Main text: On Pages 6

These designs work in concert, delivering exceptional low-field response, as evidenced by a higher magnetoresistance ratio at 10 mT relative to all printed magnetoresistive sensors reported previously^{21–31,40} (Fig. 1h and Supplementary Table 1).

Comment 3: 3. In Figure 1h, the authors highlight the large-scale fabrication, green synthesis, biocompatibility, biodegradability, recyclability, and %MR at 10 mT to emphasize the practical significance of their sensors. However, it is unclear which specific synthesis parameter(s) was/were considered in this comparison. Based on the figure caption, it seems only the binders were compared. The authors are requested to clarify the parameters used and justify their selection to ensure that the comparison is scientifically meaningful and is not limited to a single fabrication variable.

Response: We would like to emphasize that Fig. 1h is not intended to compare specific synthesis or processing parameters within our Fe/Fe₃O₄ system, nor to isolate a single fabrication variable such as the binder. Instead, Fig. 1h provides a system-level benchmark of previously reported printed magnetoresistive sensors from the literature, highlighting their practical relevance by combining (i) the functional performance at low field (%MR at 10 mT) with (ii) application- and sustainability-related attributes, including large-scale fabrication compatibility, green processing, biocompatibility, biodegradability, and recyclability.

Importantly, the comparison in Fig. 1h is based on the complete reported sensor systems, including both the magnetoresistive functional materials (e.g., [Co/Cu] multilayer microflakes, NiFe-based systems, Bi microparticles, Fe-based fillers) and the associated binder matrices/processing routes, as summarized quantitatively in Supplementary Table 1. We have revised the content, figure caption of Fig. 1h and the corresponding text to explicitly direct readers to Supplementary Table 1, thereby improving transparency and preventing the interpretation that the comparison is limited to a single fabrication variable.

By contrast, the influence of specific synthesis and processing parameters within our system is systematically investigated elsewhere in the manuscript, including oxidation temperature (Fig. 3e), oxidation environment (Supplementary Fig. 23), oxidation time (Supplementary Fig. 24) and particle size (Supplementary Fig. 15), particle volume ratio (Supplementary Fig. 24), binder type (Fig. 2d and Supplementary Fig. 6), and substrate type (Supplementary Figs. 7 and 8). Collectively, these data demonstrate that the reported performance originates from a carefully optimized material and structural design rather than from any single fabrication parameter.

Revision:

On pages 6

These designs work in concert, delivering exceptional low-field response, as evidenced by a higher magnetoresistance ratio at 10 mT relative to all printed magnetoresistive sensors reported previously^{21-31,40} (Fig. 1h and Supplementary Table 1).

On the content and caption of figure 1h:

h, Comparison of different printed MR sensors in terms of the large-scale and green fabrication (non-toxic solvent and energy saving), biodegradable, biocompatible, recyclable and MR ratio at 10 mT. A detailed quantitative comparison of the literature-reported sensor systems (including fabrication process, functional elements, binders, and low-field metrics) is provided in Supplementary Table 1. The corresponding functional fillers and binders in references: Ref.21: [Co/Cu]₅₀ flakes with PMMA; Ref.22:[Co/Cu]₅₀ flakes with PECH; Ref.23: FeCo particles with Hexadecylamine; Ref.24: [Py/Cu]₅₀ flakes with PSBS; Ref.25: [Co/Cu]₃₀ flakes with PECH; Ref.26: [Co/Cu]₅₀ flakes with PECH; NiCo particles with PECH; Ref.27: FeCoNi/Cu nanowire with PDMS; Ref.28: [Ta/Py] flakes with PSBS; Ref.29: NiFe nanowire with PDMS; Ref.30: NiFe particles with PDMS; Ref.31: Bi particles with Butylmethacrylate; Ref.40: Fe particles with Hexadecylamine.

Comment 4: *4. The authors use an acid wash (Vitamin C) treatment for the selective removal of the Fe₃O₄ shell and demonstrate a transition to linear current-voltage behavior, indicative of ohmic conduction, whereas the printed Fe/Fe₃O₄ samples exhibit nonlinear current-voltage characteristics, which they attribute to the variable-range hopping mechanism (Supplementary Fig. 5). In addition, on page 9, the authors claim that “The magnetoresistive behavior of Fe/Fe₃O₄ core-shell particle sensors can be attributed to spin-dependent hopping across magnetically disordered interfaces between adjacent Fe₃O₄ shells. Upon application of an external magnetic field, the alignment of local magnetic moments reduces spin disorder and enhances carrier hopping mobility, giving rise to negative magnetoresistance effects [Ref. 47,48].” Authors are requested to include the relevant MR curve (if possible) for a direct and robust visual demonstration of the indispensable role of the magnetically disordered interfaces between adjacent Fe₃O₄ shell in generating the magnetoresistance.*

Response: In our work, the magnetoresistance is attributed to spin-dependent hopping across magnetically disordered interfaces between adjacent Fe₃O₄ shells, as discussed in the section “*Spin-dependent hopping-mediated grain boundary magnetoresistance ...*”. Consistent with this mechanism, the printed Fe/Fe₃O₄ composites exhibit nonlinear I–V behavior (Fig. 3a and Supplementary Fig. 9) and negative temperature-dependent resistance (Fig. 3b and Supplementary Fig. 11) characteristic of hopping-type transport, whereas selective removal/reduction of the oxide shell by the Vitamin-C (acid wash) treatment leads to a transition to linear (ohmic) I–V behavior (Supplementary Fig. 9) and positive (metal) temperature-dependent resistance behavior (Supplementary Fig. 11), indicating that charge transport becomes dominated by the metallic Fe core rather than interfacial hopping.

Importantly, the MR response is strongly suppressed when the oxide shell is destroyed and removed: the acid-washed (shell-removed) samples show only a very small low-field MR (e.g., MR at 10 mT \approx 0.04%, Supplementary Table 2), close to the Fe-dominated baseline. Moreover, the MR decreases with increasing bias voltage (Fig. 3d and Supplementary Fig. 10), which is consistent with a bias-induced reduction of the interfacial hopping barrier and the associated suppression of spin-dependent transport.

To address the Reviewer’s request for a more direct visualization, we have added a representative MR(H) curve of the Vitamin-C treated (shell-removed) sample together with the sample printed with pristine particles and the sample printed with optimal oxidized particles (new Supplementary Fig. 25). This direct comparison clearly shows that the pronounced MR arises only when Fe₃O₄–Fe₃O₄ interfaces are present.

Finally, we note that the Fe core primarily contributes to magnetic flux concentration rather than being the origin of MR: after removing an average gain factor (\sim 10), the normalized MR curve of Fe/Fe₃O₄ closely resembles that of the printed Fe₃O₄ counterpart (Fig. 4a and Supplementary Fig. 15a), supporting

that the MR originates from the Fe_3O_4 interfacial grain-boundary network while the Fe core amplifies the local field at these interfaces.

Supplementary Fig. 25. a. MR curves of printed Fe/Fe₃O₄ sensor post-treated by vitamin C solution with different concentration. **b.** MR curves of printed Fe/Fe₃O₄ sensor with optimally oxidized particles, pristine particles (naturally oxidized) and shell removed particles.

Comment 5: 5. The authors claimed an improved low-field sensitivity with exceptionally reliable practical functionality. However, a comparison of devices produced within the same batch and different batches would strengthen the scientific rigor of the technology. The authors are requested to provide a statistical analysis of the key performance parameters based on measurements from multiple independent sensor devices to ensure reproducibility. The Methods section should explicitly state the sample size (n) for each dataset.

Response: To demonstrate reproducibility, we fabricated and measured three independent batches of Fe/Fe₃O₄ sensors prepared under identical conditions, as mentioned in Methods. Each batch contains $n = 10$ independent sensors (total $n = 30$), and we performed a statistical analysis of the key performance metrics, including MR at 10 mT, saturated MR at 100 mT, and maximum sensitivity. The results show good within-batch consistency and high batch-to-batch reproducibility. Specifically, the batch-wise averages are:

Batch 1 ($n=10$): MR@10 mT = $1.74\% \pm 0.14\%$; MR@100 mT = $3.00\% \pm 0.18\%$; $S_{\max} = 3.98 \pm 0.33 \text{ T}^{-1}$

Batch 2 ($n=10$): MR@10 mT = $1.67\% \pm 0.13\%$; MR@100 mT = $2.98\% \pm 0.09\%$; $S_{\max} = 3.82 \pm 0.25 \text{ T}^{-1}$

Batch 3 ($n=10$): MR@10 mT = $1.96\% \pm 0.08\%$; MR@100 mT = $3.21\% \pm 0.14\%$; $S_{\max} = 4.37 \pm 0.19 \text{ T}^{-1}$

All devices ($n=30$): MR@10 mT = $1.79\% \pm 0.17\%$; MR@100 mT = $3.06\% \pm 0.18\%$; $S_{\max} = 4.05 \pm 0.33 \text{ T}^{-1}$

The corresponding coefficients of variation (CV) remain low (typically $\sim 3\text{--}10\%$ across these metrics), confirming robust reproducibility. We have summarized these statistics in a new supplementary table 4. and visualized the within-/between-batch distributions using statistic box plots (Supplementary Fig. 3). Finally, following the reviewer's request, we have updated the Methods section to explicitly state the sample size (n) for each dataset and add the reproducibility discussion on page 6.

Revision:

Added new Supplementary Fig. 3.

Supplementary Fig. 3. Within-batch and batch-to-batch reproducibility of printed Fe/Fe₃O₄ sensors. **a.** Magnetoresistance at 10 mT (MR@10 mT) for three independently fabricated batches (Batch 1–3, n = 10 devices per batch) and the entire dataset (all samples, n = 30). **b.** Saturated magnetoresistance at 100 mT (MR@100 mT) for the same samples. **c.** Maximum low-field sensitivity (S_{\max}) for the same samples. In all panels, box plots indicate the median (center line), interquartile range (box), and 1.5×IQR whiskers; individual data points represent measurements from independent samples.

Added new Supplementary Table 4.

Supplementary table 4. Device-to-device and batch-to-batch reproducibility of printed Fe/Fe₃O₄ sensors. Statistical summary of key performance parameters measured for three independently fabricated batches (Batch 1–3, n = 10 devices per batch; total n = 30): magnetoresistance at 10 mT (MR@10 mT), saturated magnetoresistance at 100 mT (MR@100 mT), and maximum sensitivity (S_{\max}). For each batch and for the pooled dataset, the mean, standard deviation (SD), and coefficient of variation (CV) are reported.

Batch Number	Sample number	MR at 10 mT	Saturated MR (at 100 mT)	Max Sensitivity (T^{-1})
Batch #1	#1	2.0%	3.2%	4.52
	#2	1.7%	2.8%	3.96
	#3	1.6%	2.9%	3.69
	#4	1.7%	2.9%	3.96
	#5	1.8%	2.9%	4.05
	#6	1.7%	3.1%	3.85
	#7	1.5%	2.9%	3.52
	#8	1.7%	2.9%	3.92
	#9	1.8%	3.0%	3.98
	#10	1.9%	3.4%	4.31
	Mean	1.74%	3.00%	3.98
	SD	0.14%	0.18%	0.33
CV	8.05%	6.00%	8.29%	
Batch #2	#1	1.6%	2.9%	3.61
	#2	1.6%	3.1%	3.55
	#3	1.6%	2.8%	3.65
	#4	1.6%	2.9%	3.72
	#5	1.9%	3.0%	4.27
	#6	1.8%	3.0%	4.12
	#7	1.5%	3.0%	3.57
	#8	1.8%	3.1%	4.04
	#9	1.7%	3.0%	3.88

	#10	1.6%	3.0%	3.81
	Mean	1.67%	2.98%	3.82
	SD	0.13%	0.09%	0.25
	CV	7.78%	3.02%	6.54%
Batch #3	#1	1.9%	3.1%	4.32
	#2	1.9%	3.1%	4.22
	#3	1.9%	3.1%	4.24
	#4	1.9%	3.2%	4.18
	#5	1.9%	3.2%	4.17
	#6	2.1%	3.4%	4.62
	#7	2.1%	3.4%	4.75
	#8	2.0%	3.4%	4.47
	#9	2.0%	3.0%	4.39
	#10	1.9%	3.2%	4.29
	Mean	1.96%	3.21%	4.37
	SD	0.08%	0.14%	0.19
	CV	4.08%	4.36%	4.35%
All	Mean	1.79%	3.06%	4.05
	SD	0.17%	0.18%	0.33
	CV	9.50%	5.88%	8.15%

Main text: page 5:

The printed sensors also exhibit robust device-to-device performance reproducibility (Supplementary Fig. 3 and Supplementary Table 4).

Method: In the “Magnetoresistance measurement” section. On page 18.

Added:

To assess performance reproducibility, sensors from three independently fabricated batches were measured (batch 1-3, n = 10 devices per batch; total n = 30), prepared under identical conditions.

Comment 6: 6. On page 11, the authors have successfully demonstrated the optimization of device performance by varying the core diameter and shell thickness using numerical simulations. The authors also mentioned “..... as the core diameter increases, the greater permeability contrast improves flux concentration. However, beyond a critical size, the resulting flux distribution reaches a geometric limit, no longer enhancing the field in grain boundaries.” The observed normalized MR shown in Figure 4i is related to the overall particle size. The authors are required to provide direct microstructural evidence (e.g., cross-sectional SEM/TEM/HRTEM) to quantify the final core diameter and precise shell thickness of the particles. These values should be explicitly stated in the main text and correlated with the performance data.

Response: We agree that quantifying and explicitly reporting the final core diameter and shell thickness, and relating these values to the gain-factor trend in Fig. 4i, will make the structure-performance interpretation clearer and more rigorous. In Fig. 4i, we investigate how the gain factor evolves with particle (core) size. Accordingly, we prepared three representative particle size classes with average diameters of ~90 nm, ~3 μm, and ~15 μm, quantified by statistical size analysis (Supplementary Figs.

19 and 20). All three size classes were oxidized under the same conditions (235 °C, medium vacuum 0.5 mbar, 30 min) to keep the oxide-shell formation comparable across samples (mentioned in Methods). Direct microstructural quantification of the shell thickness under the optimized oxidation conditions (235 °C, medium vacuum 0.5 mbar, 30 min) is provided by cross-sectional TEM/HR-TEM of the representative $\sim 3 \mu\text{m}$ Fe/Fe₃O₄ core-shell particles (Fig. 3f and Supplementary Fig. 22), showing a continuous Fe₃O₄ shell with a thickness of about 10-20 nm. For the core-size series, the use of identical oxidation conditions is further supported by literature indicating that, under the same oxidation conditions, the oxide growth kinetics (and thus shell-thickness evolution) are largely independent of particle size for iron/iron-oxide core-shell formation (DOI: 10.1103/PhysRevB.68.195423). Therefore, the experimentally observed evolution and saturation of the normalized MR/gain factor with increasing particle size (Fig. 4i and j) provide direct experimental support for the simulation-predicted trend: increasing core diameter enhances flux concentration until a geometric limit is reached.

To improve clarity, we have now explicitly stated the particle diameters (90 nm / 3 μm / 15 μm), and the unified oxidation condition (235 °C, 0.5 mbar, 30 min) in the caption of Fig. 4.

In addition, we separately examined shell-growth-related effects experimentally by varying oxidation time at fixed temperature/vacuum (Fig. 4f, g and Supplementary Fig. 15b), which shows how thicker shells reduce the gain factor and modify the low-field response. Together, these results provide a clear and experimentally grounded link between (i) the simulated trends and (ii) the measured gain-factor evolution: core diameter controls the flux-guiding enhancement observed in Fig. 4i, while shell thickness is validated independently via controlled oxidation-time variations for particles of the same size (3 μm).

Revision:

Caption of Figure 4.

h, SEM images of Fe/Fe₃O₄ core-shell particles with different sizes (90 nm, 3 μm and 15 μm), Scale bars: 500 nm, 10 μm , 100 μm from top to bottom. For the distribution of particle diameters, see Supplementary Figs. 13 and 14. All three size classes were oxidized under the same conditions (235 °C, medium vacuum 0.5 mbar, 30 min) to keep the oxide-shell formation comparable across samples. i, Normalized MR curves and j, sensitivity curves of printed Fe/Fe₃O₄ core-shell particles with different sizes.

Comment 7: 7. The claim of a "truly closed-loop sustainable life cycle" (e.g., Page 12, Fig. 5a) should be reconsidered. This phrase implies a rigorous Life Cycle Assessment (LCA) that quantitatively evaluates environmental impacts from raw material extraction to end-of-life. The authors are encouraged to reflect on the presented evidence more accurately. For instance, wording such as "demonstrates key attributes for a sustainable life cycle" or "presents a viable pathway towards closed-loop sustainability" would be more appropriate. Moreover, including a qualitative discussion acknowledging that a comprehensive quantitative LCA would be a valuable future direction could add important nuances and strengthen the environmental relevance of this study.

Response: We have carefully reconsidered and refined our wording to ensure that the environmental claims are aligned with the evidence presented in this study, and we fully agree that a comprehensive quantitative life cycle assessment (LCA) would be a valuable future direction to rigorously evaluate cradle-to-grave environmental impacts.

We have therefore revised the wording throughout the manuscript (Abstract, Introduction, Application section, Discussion, and the Fig. 5 caption) to more accurately reflect the evidence presented in this

work. In addition, following the reviewer's suggestion, we have added a qualitative outlook in the Discussion explicitly stating that future work would benefit from a comprehensive quantitative LCA to quantify cradle-to-grave impacts and guide continued optimization.

Revision (for comment 7):

1. Abstract. On page 2.

~~“These properties collectively ensure full lifecycle eco-sustainability.”~~

changed to

“These properties collectively demonstrate key attributes for a sustainable life cycle.”

2. Introduction. On page 4.

~~“In this work, we successfully incorporate environmental sustainability into the entire lifecycle of printable magnetoresistive sensors while equipping them with enhanced low-field sensing performance.”~~

changed to

“In this work, we integrate environmental sustainability considerations across key lifecycle stages of printable magnetoresistive sensors while equipping them with enhanced low-field sensing performance.”

3. Application section. On page 12.

~~“enables a truly closed-loop sustainable life cycle”~~

changed to

“presents a viable pathway toward closed-loop sustainability”

4. Fig. 5 caption.

~~“Conceptual illustration showing the closed-loop eco-sustainability of the biodegradable MR sensors.”~~

changed to

“Conceptual illustration showing the pathway toward closed-loop eco-sustainability of the biodegradable MR sensors.”

5. Discussion

We add “Future work would benefit from a comprehensive quantitative life cycle assessment (LCA) to further quantify cradle-to-grave impacts and guide continued optimization.”

Minor comment 1: *i) The placement of Figure 1c in the current version may be confusing for readers. The authors are encouraged to rearrange the figure to ensure a more logical flow and clearer distinction from Figures 1d and 1e. Adding labels such as “Fe core (conductor & flux guide)” and “Fe₃O₄ shell (MR-active)” to the schematic would make the synergistic mechanism immediately clearer.*

Response: We have revised the layout and labeling of Fig. 1 accordingly.

Revision:

Minor comment 2: ii) In Supplementary figure 11, the colors described in the caption do not correspond to those of the respective curves in the graph, particularly the green curve in S11-a. The authors are requested to verify this inconsistency.

Response: We have verified the issue and corrected the color descriptions in the caption of Supplementary Fig. 11 (new Supplementary Fig. 15) so that they now match the corresponding curves. In addition, we also updated the related color references in Fig. 4 accordingly.

Revision: New Supplementary Fig. 15 caption.

Supplementary Fig. 15. MR curve comparison. a, MR curves of printed Fe/Fe₃O₄ core-shell particles (cyan curve), with which removed 10 gain factor (purple curve) and printed Fe₃O₄ particles (red curve).

b, MR curves of printed Fe/Fe₃O₄ core-shell particles with different time (30 min, 60 min, 90 min, and 120 min). c, MR curves of printed Fe/Fe₃O₄ core-shell particles with different size.

Fig. 4 caption.

a, Normalized MR curves and **b**, sensitivity curves comparison of printed Fe/Fe₃O₄ core-shell particles (cyan curve), with which removed 10 gain factor (purple dashed line curve) and printed Fe₃O₄ particles (red curve).

Minor comment 3: *iii) Page 8, section “Spin-dependent hopping-mediated grain boundary” – The sentence “..., the Fe/Fe₃O₄ sensors exhibit resistance reduction by over two orders of magnitude...” should cite the specific supplementary table where these data are presented (apparently referring to Supplementary Tables 3). The authors are advised to verify this reference.*

Response: We have verified the sentence to explicitly reference Supplementary Table 3, which contains the resistance values supporting this statement.

Revision:

Main text: On page 9.

Compared with composites printed with Fe₃O₄ microparticles, the Fe/Fe₃O₄ sensors exhibit resistance reduction by over two orders of magnitude (Supplementary Table 3), rendering them suitable for practical electronic applications.

Reviewer #3 (Remarks to the Author):

Response: We sincerely thank Reviewer #3 for co-reviewing our manuscript as part of the Nature Communications initiative to support peer-review training and to recognize Early Career Researchers.